# Multi-layer Retrieval of Aerosol Optical Depth in the Troposphere Using SEVIRI Data, Case study: European Continent

Maryam Pashayi[1], Mehran Satari[1], Mehdi Momeni Shahraki[1]

[1]Department of Geomatics Engineering, Faculty of Civil Engineering and Transportation, University of Isfahan, Isfahan, I817467344, 1ran

*Correspondence to*: Mehran Satari (sattari@eng.ui.ac.ir)

**Abstract.** Multi-layer Aerosol Optical Depth (AOD) estimation with sufficient spatial and temporal resolution is crucial for effective aerosol monitoring, given the significant variations over time and space. While ground-based observations provide detailed vertical profiles, satellite data are essential for addressing the spatial and temporal gaps. This study utilizes profiles from the Cloud-Aerosol Lidar with Orthogonal Polarization (CALIOP) and data from the Spinning Enhanced Visible and Infrared Imager (SEVIRI) to estimate vertical AOD values at 1.5, 3, 5, and 10 km layers. These estimations are achieved with spatial and temporal resolutions of 3 km × 3 km and 15 minutes, respectively, over Europe troposphere. We employed machine learning models—XGBoost (XGB) and Random Forest (RF)—trained on SEVIRI data from 2017 and 2018 for the estimations. Validation using CALIOP AOD retrievals in 2019 confirmed the reliability of our findings, emphasizing the importance of wind speed (Ws) and wind direction (Wd) in improving AOD estimation accuracy. A comparison between seasonal and annual models revealed slight variations in accuracy, leading to the selection of annual models as the preferred approach for estimating SEVIRI multi-layer AOD values. Among the annual models, the XGB model demonstrated superior performance over the RF model at all four layers, yielding more reliable AOD estimations with R² values 0.99, 0.97, 0.98, and 0.98 for the four layers from low to high altitude layers. Further validation using data from EARLINET stations across Europe in 2020 indicated that the XGB model still achieved better agreement with EARLINET AOD profiles, with R² values of 0.86, 0.80, 0.75, and 0.59, and RMSE values of 0.022, 0.012, 0.015, and 0.005, respectively. We performed a qualitative validation of multi-layer AOD estimations by comparing spatial trends with CALIOP AOD retrievals for SEVIRI pixels on four dates in 2019, showing strong agreement across varying AOD levels. Additionally, the model successfully estimated AOD at 15-minute intervals for two real events—a Saharan dust plume and the Mount Etna eruption—revealing consistent physical characteristics, including long-range transport in the upper layers and a gradual increase in AOD from lower to higher tropospheric layers during volcanic events. The results demonstrate that the proposed method facilitates comprehensive monitoring of AOD behaviour throughout the four vertical layers of the troposphere, offering important insights into the dynamics of aerosol occurrence.

**Keywords:** AOD Vertical Profile, SEVIRI, Geostationary satellite, CALIOP, EARLINET, Machine Learning.

## 1 Introduction

Aerosols are recognized as significant contributors to air pollution, climate change, and the modification of solar and thermal infrared radiation absorption and scattering (Hyslop, 2009; Pope et al., 2019; Li et al., 2022). Understanding aerosol behaviour in the troposphere is vital for enhancing atmospheric models and refining monitoring techniques. Aerosol Optical Depth (AOD) is a critical parameter for quantitatively estimating aerosol concentration and its optical properties. Recent research emphasizes the importance of multi-layer retrieval of AOD in reducing uncertainties associated with aerosol characterization (Wang et al., 2018; Rogozovsky et al., 2021; Gupta et al., 2021; Rogozovsky et al., 2023). Additionally, an in-depth investigation of the multi-layer distribution of aerosol properties within the troposphere is essential for elucidating aerosol transport mechanisms, facilitating source identification, and improving atmospheric dynamics models. This understanding ultimately enhances the accuracy of simulations related to long-range aerosol transport (Chen et al., 2023). Vertical AOD retrieval can be conducted through ground-based observations or inferred from remote sensing data. Ground-based LiDAR networks, like the European Aerosol Research Lidar Network (EARLINET), provide detailed insights into aerosol characteristics by offering vertical profiles of optical properties, enabling high-resolution, multi-layer AOD retrieval through precise quantification of aerosol loading across distinct atmospheric layers (Bösenberg et al., 2001, 2003). While these observations offer detailed vertical information, their sparse nature necessitates supplementation with satellite observations. Satellite LiDAR remote sensing emerges as the primary method for capturing global temporal and spatial variations in aerosol profiles. The Cloud-Aerosol Lidar with Orthogonal Polarization (CALIOP) onboard the Cloud-Aerosol Lidar and Infrared Pathfinder Satellite Observation (CALIPSO) satellite launched in 2006, offer distributions of aerosols and clouds, along with their geometrical and optical properties. Multi-layer AOD values are retrieved using the level 2 aerosol extinction profiles at both 532 nm and 1064 nm, where the aerosol extinction profiles are determined from backscatter measurements (Winker et al., 2004, 2006, 2007). However, the CALIOP sensor encounters challenges in achieving adequate spatial and temporal coverage, with limitations in daily and global resolution (16-day temporal resolution and 5 km profile distance).

Recent advancements have sought to overcome these limitations through the use of passive satellite sensors with varying temporal resolutions, such as the Tropospheric Monitoring Instrument (TROPOMI), which provide near-daily global coverage with a spatial resolution 3.5 × 7 km (improved 3.5 × 5.5 km in 2019) and was launched in 2017 on Sentinel-5P satellite (Veefkind et al., 2012); the Earth Polychromatic Imaging Camera (EPIC), offering a continuous daytime view every 60 to 100 minutes with a spatial resolution of about 8x8km since its launch on February 11, 2015, onboard the Deep Space Climate Observatory (DSCOVR) satellite (Marshak & Knyazikhin, 2017); the Global Ozone Monitoring Experiment-2 (GOME-2) on Meteorological Operational Satellite Program (MetOp-C), with a three-day revisit cycle and a spatial resolution of approximately 40 × 40 km since 2018; and the Moderate Resolution Imaging Spectroradiometer (MODIS), onboard Terra (launched in 1999) and Aqua (launched in 2002), provides daily global coverage with spatial resolutions ranging from 0.25 to 1 km (Lyapustin et al., 2011).

The relevant researches focus on various methods specifically aimed at retrieving aerosol layer height (ALH) rather than AOD at different altitudes. One prominent method, Oxygen ($O_2$) A and B band Absorption Spectroscopy, utilizes the differential absorption of sunlight by $O_2$ molecules at different altitudes (Zeng et al., 2018; Xu et al., 2017; Xu et al., 2019). Elevated aerosol layers scatter sunlight back to space, shortening the atmospheric path length and decreasing $O_2$ absorption. By analyzing spectral characteristics in the $O_2$ A and B bands, researchers infer ALH. However, retrieval sensitivity is enhanced over darker surfaces and higher AOD, making it challenging over bright surfaces or under low aerosol loading. For instance, Nanda et al. (2020) employed TROPOMI observations with an optimal estimation scheme in the $O_2$ A band, assuming a uniformly distributed aerosol layer. Similarly, the algorithm developed using EPIC/DSCOVR data leverages atmospheric window bands and Differential Optical Absorption Spectroscopy (DOAS) ratios, integrating MODIS and GOME-2 surface reflectance data. For retrievals over vegetated areas, the algorithm favours the $O_2$ B band due to its lower surface reflectance (Xu et al., 2019). Another study combined $O_2$ A and B band data from Scanning Imaging Absorption Spectrometer for Atmospheric Chartography (SCIAMACHY) and GOME-2 for enhanced ALH sensitivity, especially near boundary layers (Hollstein & Fischer, 2014).

Additional retrieval method, Stereoscopic techniques—employed by the Multi-angle Imaging SpectroRadiometer (MISR), launched in 2000— utilize multi-angle observations to geometrically determine plume heights. MISR offers a spatial resolution of approximately 275 meters and a temporal resolution of around once every 7 days, making it especially useful over reflective surfaces, as it relies on geometric data rather than surface reflectance (Muller et al., 2002; Zakšek et al., 2013; Fisher et al., 2014; Val Martin et al., 2018).

Passive satellite-based ALH retrieval techniques, while offering global coverage, often simplify the aerosol vertical distribution by assuming a single homogeneous layer (Zeng et al., 2018; Xu et al., 2017; Xu et al., 2019). This simplification can lead to inaccurate representations of complex aerosol profiles, especially in cases of multi-layered events. In addition, these passive satellite-based methods face further constraints due to the low spatial resolution of instruments like EPIC and GOME-2, as well as low temporal resolution of sensors such as TROPOMI, GOME-2, and MISR. These constrains on resolution reduce the effectiveness of these retrievals in capturing fine-scale, rapidly evolving aerosol distribution events, such as smoke plumes from fires.

Other studies by Pashayi et al., (2023, 2024) have introduced Seasonal and Seasonal-Independent machine learning models for AOD retrieving in multiple layers. These models seek to investigate the relationship between MODIS observations and CALIOP AOD for retrieval multiple layers AOD values at a spatial-temporal resolution corresponding to the MODIS AOD product. This analysis focuses specifically on the Persian Gulf region. The researchers subsequently analyse their findings using CALIOP AOD retrievals across multiple vertical layers. Although these studies have advanced the retrieval of AOD across multiple layers, the constraint of MODIS's daily temporal resolution remains a significant limitation (Wei et al., 2020). Geostationary satellites, such as Himawari-8 (launched in 2014), which is equipped with the Advanced Himawari Imager (AHI; Da, 2015), the Advanced Baseline Imager (ABI; Kalluri et al., 2015) onboard the Geostationary Operational Environmental Satellite (GOES; launched in 2016), and the Meteosat geostationary satellites featuring the Spinning Enhanced

Visible and Infrared Imager (SEVIRI; launched in 2002; Pasternak et al., 1994), provide sub-hourly, high-resolution observations that significantly enhance global aerosol monitoring capabilities across diverse regions (Schmit et al., 2018; Zhang et al., 2019; Ge et al., 2019; Tang et al., 2019; Zawadzka-Manko et al., 2020; Witthuhn et al., 2020; Kocaman et al., 2022; Ceamanos et al., 2023). Notably, SEVIRI offers high temporal and spatial resolutions, presenting valuable opportunities

to expand aerosol datasets for Europe (Stebel et al., 2021; Nicolae et al., 2021). Consequently, utilizing observations from these satellites enables the multi-layer retrieval of AOD, effectively addressing the limitations associated with temporal and spatial resolution presented in the previous studies.

The retrieval of multi-layer AOD values from passive satellites observations typically entails two primary approaches: physically based (Seidel et al., 2012; Lipponen et al., 2018; Amini et al., 2021; Mehta et al., 2022) and data mining approaches

(Radosavljevic et al., 2010; She et al., 2020; Chen et al., 2022). The physically based approach relies on established principles of aerosol behaviour, utilizing models derived from physical laws to retrieve AOD values. This approach often involves simplifications and assumptions, such as treating the atmosphere as a single aerosol layer in most of the passive satellite-based ALH retrieval algorithms previously mentioned. While this assumption is necessary for practical implementation, it can introduce uncertainties and limit the accuracy of retrievals, particularly in complex scenes. Additionally, physical based

methods are sensitive to surface reflectance. Over bright surfaces, the contribution of surface reflection to the top-of-atmosphere (TOA) radiance can dominate, making it challenging to extract a clear aerosol signal, especially for low aerosol loading. This limitation underscores the need for accurate surface reflectance characterization (Xu et al., 2017; Xu et al., 2019; Nanda et al., 2020).

In contrast, data mining approaches offer a promising alternative by harnessing large datasets and employing learning-based

algorithms to discern patterns and relationships within complex aerosol systems. Machine learning methods as data mining approach have the potential to learn complex relationships between AOD and other atmospheric and surface variables. These methods can capture non-linear dependencies and potentially handle multi-layer scenarios more effectively than physical models. This advantage is particularly relevant for AOD retrieval in diverse and heterogeneous environments. Additionally, integrating data from multiple sources, such as mereological, land-cover, temporal and location data, can provide more

comprehensive information for a time- and location-based AOD retrieval, particularly for multi-layer scenarios (Chen et al., 2020; Lee et al., 2022; Berhane et al., 2024).

In this study, we introduce a model for sub-hourly multi-layer AOD retrieval over Europe continent troposphere by integrating SEVIRI-based information with CALIOP aerosol profile products. To achieve this, two well-established machine learning models—XGBoost (XGB) and Random Forest (RF)— utilized for retrieving AOD values in four distinct layers, approximately

every 15 minutes, with a spatial resolution of 3 km × 3 km. The four tropospheric layers analyzed in this study are 0–1.5 km, 1.5–3 km, 3–5 km, and 5–10 km, denoted as $AOD_{1.5}$, $AOD_3$, $AOD_5$, and $AOD_{10}$, respectively. The selection of these layers for multi-layer AOD retrieval is based on the distinct aerosol transport mechanisms observed at these altitudes. The 0-1.5 km layer captures aerosols from local sources transported upwards by updrafts from the cloud base, a process called pumping. The 1.5-3 km layer, where thermal bubbles often initiate, allows examination of aerosols, potentially from mid-range sources, that are

lifted into the cloud with the rising bubble. The 3-5 km layer captures aerosols transported over longer distances that enter the cloud through entrainment at the cloud edges as the bubble ascends. The 5-10 km layer is designed to capture the influence of long-range transported aerosols on cloud properties at higher altitudes. This multi-layer approach enables analysis of how local to long-range aerosol transport contributes to aerosol-cloud interactions (Zhang et al., 2021; Lebo, 2014; Marinescu et al., 2017).

To train and validate two machine learning models, we employed AOD data retrieved from CALIOP aerosol product and EARLINET stations distributed across Europe. Model performance was qualitatively evaluated by analyzing its response to two notable aerosol events: a significant dust intrusion from March 13 to March 18, 2022, and a volcanic eruption on August 14, 2023. These events offered valuable case studies to assess the model's capability in detecting and characterizing distinct aerosol signatures across these four layers. We organized the rest of the paper as follows: Section 2 provides a comprehensive overview of the dataset employed, while Section 3 details the necessary pre-processing steps and retrieval methodology. Subsequently, Section 4 delves into the discussion of the vertically retrieved AOD results, followed by conclusions outlined in Section 5.

## 2 Study Area and Data Source

### 2.1 Study Area

The study area encompasses a significant portion of Europe troposphere, spanning from 35°N to 71°N and -7°E to 70°E, covering approximately 10.18 million square kilometres. Despite its relatively small land area, Europe exhibits a diverse geographical landscape and complex atmospheric dynamics. Urban centers in Europe face persistent air pollution issues due to industrial activities and vehicular emissions, compounded by the effects of climate change. Various aerosol types, originating from industrial processes, transportation, biomass burning, and natural events significantly impact air quality, weather patterns, and climate dynamics across the continent. Long-range transport of aerosols, particularly from sources in Africa such as Saharan dust storms, underscores the interconnectedness of atmospheric processes across continents and emphasizes the necessity of international cooperation in addressing air pollution and environmental challenges.

### 2.2 Data Source

### 2.2.1 SEVIRI

The Meteosat Second Generation (MSG) constitutes a series of four satellites managed by the Exploitation of Meteorological Satellites (EUMETSAT) and has been operational since 2004. Originally designated as MSG1 to MSG4, these satellites were subsequently rebranded as Meteosat-8 to Meteosat-11, respectively. The primary instrument onboard these satellites is the Spinning Enhanced Visible and Infrared Imager (SEVIRI), a radiometer equipped with 11 spectral channels spanning the visible to the infrared spectrum. These include the Visible (0.6 μm and 0.8 μm) channels, as well as a Near Infrared (1.6 μm) channel, provides spatial resolution of about 3 km at the sub-satellite point, and a high-resolution visible (HRV) channel

offering a finer spatial resolution of 1 km at nadir. Strategically centered at various wavelengths, the thermal channels of SEVIRI include 6.2 and 7.3 µm (targeting strong water vapor absorption), 8.7, 10.8, and 12.0 µm (window channels), as well as 9.7 µm (for ozone absorption) and 13.4 µm (for carbon dioxide absorption). This operational system delivers full-disk Earth data, while the rapid scan service focuses on observing the upper part of the Earth's disk, covering Europe and North Africa, with a repetition time of 15 minutes (Schmetz et al., 2002; Zawadzka et al., 2014). In our study, we primarily utilize SEVIRI data from Meteosat-11, the fourth and final flight unit of the MSG program, which was launched on July 15, 2015. Meteosat-11 currently operates in geostationary orbit, positioned at 36,000 km above the equator. Its coverage extends over Europe, Africa, and the Indian Ocean, spanning from -81 to 81 degrees longitude and -79 to 79 degrees latitude. Figure 1 provides a visualization of the coverage area of SEVIRI.

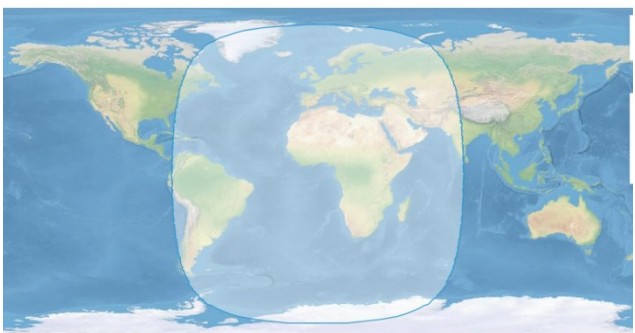

**Figure 1. The area covered by the SEVIRI instrument
(https://data.eumetsat.int/data/map/EO:EUM:DAT:MSG:HRSEVIRI).**

## 2.2.2 CALIOP

The CALIOP instrument plays a pivotal role in the CALIPSO satellite, launched in April 2006 with the primary objective of reliably delivering high-resolution vertical profiles of global aerosol properties via an active sensing technique. Functioning as a polarization-sensitive LiDAR, CALIOP measures the depolarization ratio, serving as a degree of particle irregularity. CALIOP is specifically designed to observe aerosol optical properties during both the day and night, focusing on vertical layers at wavelengths of 532 and 1064 nm. Its Level 2 algorithm not only provides information on aerosol optical characteristics like particle depolarization ratio and color ratio but also retrieves extinction coefficients. Notably, CALIOP data offer a temporal resolution of approximately 16 days, capturing insights into aerosol dynamics over time. Sampling occurs at intervals of 333 m along the orbital track, maintaining a vertical resolution of 60 m from altitudes of -0.5 to 20 km and 180 m from 20 to 30 km within the vertical profile (Winker et al., 2004, 2006, and 2007). For this study, we employed CALIOP level 2 Version 4.2 aerosol profile products from 2017 to 2019 to estimate multi-layer AOD values within the defined study region.

### 2.2.3 MODIS land cover data

In this research, we leveraged land cover data spanning 2017 to 2019, with a spatial resolution of 1 km, sourced from the global MODIS products (MCD12Q1 V6) covering Europe. These data, derived from both Terra and Aqua satellites, provide comprehensive land cover types annually from 2001. The dataset encompasses six classification schemes, elucidated in the downloadable User Guide available at https://ladsweb.modaps.eosdis.nasa.gov/. Each MCD12Q1 Version 6 Hierarchical Data Format 4 (HDF4) file comprises layers for Land Cover Type 1-5, Land Cover Property 1-3, Land Cover Property Assessment 1-3, Land Cover Quality Control (QC), and a Land Water Mask (Sulla-Menashe and Friedl, 2018). Our study specifically focuses on the first classification scheme, the Annual International Geosphere-Biosphere Program (IGBP) classification.

### 2.2.4 Meteorological data

Meteorological data were acquired from the European Centre for Medium-Range Weather Forecasts (ECMWF) dataset, accessible at (https://cds.climate.copernicus.eu/). ECMWF has been actively operational in real-time seasonal forecast systems since 1997, providing access to standard meteorological data. This dataset comprises two distinct sets of data (Copernicus Climate Change Service, Climate Data Store, (2021)). Firstly, version 2 of the Integrated Global Radiosonde Archive (IGRA) from 1978 integrates global radio sounding containing temperature, humidity, and wind data from various sources. The dataset is presented in the form of a global grid with a conventional grid resolution of 0.25° × 0.25°. Compared with previous-generation products, the temporal resolution has been increased from 6 hours to 1 hour, enabling the study of diurnal variations in the troposphere. Secondly, the Radio Sounding HARMonization (RHARM) homogenized dataset offers adjusted values for temperature, relative humidity, and wind. RHARM effectively eliminates systematic effects such as variations in measurement sensors, biases induced by solar radiation, calibration drifts, station relocations, and other factors, across 700 IGRA radiosonde stations and ship-based radio soundings. RHARM includes twice-daily (0000 and 1200 UTC) radiosonde data at mandatory and standard levels, featuring essential parameters like air temperature (T, K), air pressure (P, Pa), wind speed (Ws, m/s), and wind direction (Wd, degrees from north). For this study, the global grid dataset is utilized over the European continent from 2017 to 2019, as depicted in Fig. 2.

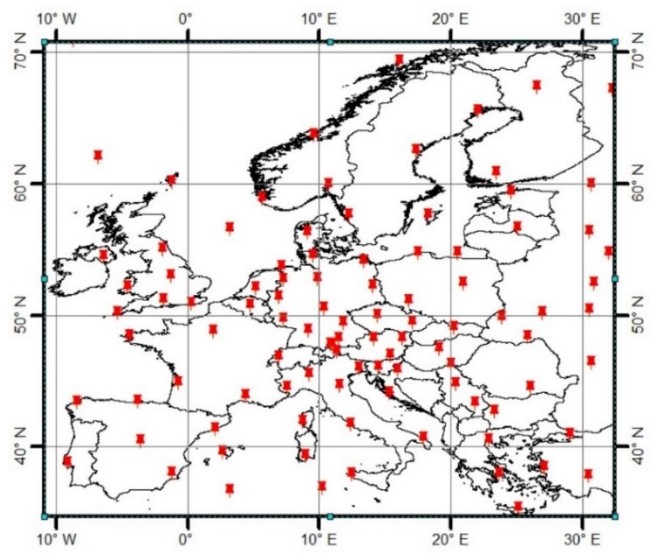

**Figure 2. Map depicting the ECMWF stations for meteorological data measurements (Durre et al., 2016).**

### 2.2.5 EARLINET

EARLINET, established in the year 2000 (Bösenberg et al., 2001, 2003), originated as a research project funded by the European Commission within the framework of the Fifth Framework Program. The primary objective of EARLINET is to generate profiles of aerosol optical properties, thereby constructing an expansive, quantitative, and statistically robust database for the continental-scale distribution of aerosols. This initiative aims to enhance network operations, facilitate research on aerosol-related processes, validate satellite sensor data, advance model development and validation, integrate aerosol data into operational models, and compile a comprehensive climatology of aerosol distribution. Currently, the network comprises 30 active stations, with the majority equipped with Raman LiDAR featuring depolarization channels. These Raman LiDAR - operating EARLINET stations typically provide profiles of aerosol extinction and backscatter coefficients without relying on significant assumptions. Figure 3 illustrates the distribution of EARLINET stations over the study area.

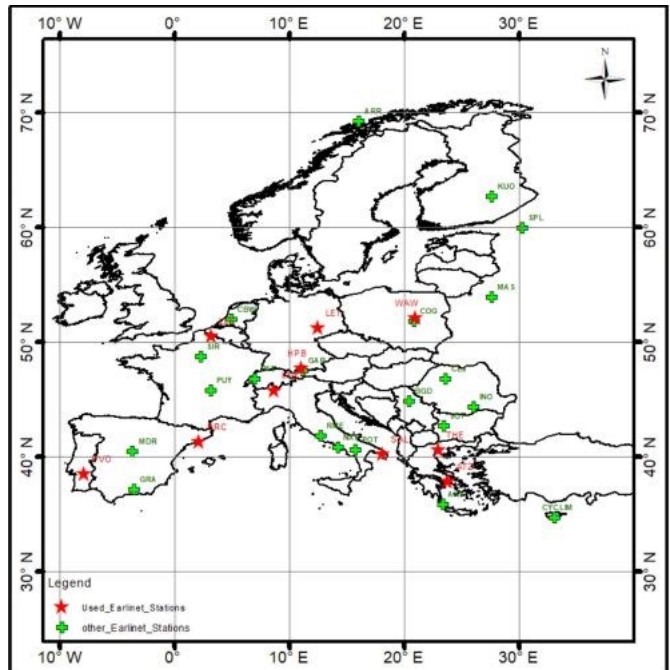

**Figure 3. Map depicting currently active EARLINET stations (https://data.earlinet.org/earlinet/). The red stars indicate the geographical distribution of EARLINET LiDAR stations used in this study.**

## 3 Methodology

As noted, hyperspectral measurements in the oxygen bands enable aerosol vertical distribution retrieval by analyzing photon path length changes due to scattering at different altitudes. SEVIRI's spectral bands, however, are primarily designed for cloud and land surface observations and do not specifically cover the oxygen bands. The SEVIRI bands closest to the oxygen bands

are $B_1$ (635 nm) and $B_2$ (810 nm), which are in the visible spectrum and respond to scattering by vertically distributed aerosols. The near-infrared and shortwave infrared (SWIR) bands ($B_3, B_4, B_7, B_9$, and $B_{10}$) are indirectly influenced by aerosol vertical distribution, as the accuracy of AOD retrievals using these bands can be affected by aerosol layering. While these bands may not directly provide vertical profile information, they could yield complementary data that, when combined with other wavelength retrievals, enhances the understanding of aerosol vertical distribution (Wu et al., 2017; Li et al., 2020). $B_5$ and $B_6$

offer insights into water vapor profiles, which can be incorporated into aerosol retrieval algorithms. By accounting for water vapor influence, these bands indirectly improve the accuracy of aerosol vertical distribution estimates. The ozone band ($B_8$) contributes to atmospheric chemistry and aerosol formation insights but does not directly reveal vertical distribution, while $B_{11}$'s lower scattering efficiency limits its sensitivity to vertical variations in aerosols for direct retrieval. As a result, SEVIRI's bands provide a range of potential avenues for studying aerosol vertical distribution, with both direct and indirect contributions.

Although SEVIRI's bands offer valuable data for meteorological observations such as cloud monitoring, surface temperature

and water vapour, its spectral design is not optimized for detailed monitoring of air quality or climate through atmospheric gases and aerosols, as is TROPOMI. Thus, physical approaches for detailed multi-layer aerosol retrieval, especially for multi-layer AOD, remain challenging with SEVIRI's current spectral configuration.

Meteorological data significantly influence the vertical distribution of aerosols, with varying impacts depending on aerosol type, transport dynamics, and atmospheric conditions. Wind speed and direction drive both horizontal and vertical aerosol transport, with higher wind speeds over oceans enhancing sea salt aerosol concentrations (Kaufman et al., 1997; Yu et al., 2006; Chin et al., 2007; Tesche et al., 2009). Temperature and pressure also play critical roles; temperature inversions inhibit vertical mixing, trapping aerosols in distinct layers, while convective activity from surface heating mixes aerosols in the boundary layer, creating a more homogeneous distribution. Stable high-pressure systems promote surface accumulation by limiting mixing, whereas low-pressure systems enhance upward transport, extending aerosol atmospheric lifetimes (Tesche et al., 2009). The complexity of these interactions suggests a significant challenge for multi-layer AOD retrieval using physical approaches, as accurate modelling requires accounting for diverse meteorological influences and variations in aerosol type, transport, and vertical distribution.

Geographical location, land cover, and temporal factors significantly influence the vertical distribution of aerosols across Europe. Coastal regions tend to have elevated sea salt aerosols due to ocean surface wind activity, while continental areas, especially in winter, experience higher anthropogenic aerosol concentrations from sources like fossil fuel combustion and industrial emissions. Additionally, the latitude and prevailing wind patterns, such as easterly winds, play a role in the long-range transport of aerosols, affecting distribution both horizontally and vertically. Land cover also contributes to these dynamics: forests emit biogenic volatile organic compounds (VOCs), which can form secondary organic aerosols, while urban and agricultural areas introduce anthropogenic aerosols from activities like traffic, industrial emissions, and fertilizer use.

Temporal variations, including seasonal and diurnal changes, further complicate aerosol distribution. For example, during winter, stable high-pressure systems trap aerosols in the planetary boundary layer (PBL), while in summer, warmer temperatures enhance photochemical activity, leading to increased ozone and sulfate concentrations. Diurnal fluctuations are also evident, particularly in urban areas, where traffic and industrial activities create peaks in anthropogenic emissions during the day.

These combined effects underscore the complexity of aerosol behaviour, emphasizing the necessity for an approach that integrates all relevant variables and effectively captures their interactions and influence on vertical aerosol distribution and multi-layer AOD retrieving. Machine learning-based methodology, capable of managing large datasets and discerning intricate relationships between these variables, presents a promising solution for accurate multi-layer AOD retrieval. Our proposed model framework for estimating AOD at the at the mentioned four distinct layers over Europe continent troposphere encompasses several sequential steps: data collection, preprocessing, partitioning, regression, and analysis of the performance of each regression model to ascertain the most accurate one, as illustrated in Fig. 4.

The process commences with data collection, detailed in the preceding section. Subsequently, preprocessing of both input and output data becomes imperative to ensure their suitability for subsequent analysis. The dataset is then partitioned into two

subsets: training and testing, a pivotal step in machine learning aimed at assessing model performance and mitigating over fitting. Following data partitioning, various model structures are proposed and developed to capture the intricate relationships within the dataset. This phase entails selecting appropriate algorithms and architectures tailored to the specific task of multi-layer AOD estimation. Finally, the performance of each model is meticulously evaluated using predefined metrics to pinpoint the most accurate and reliable model for AOD estimation across the desired vertical layers. In the subsequent sections, we will delve into a detailed examination of each step.

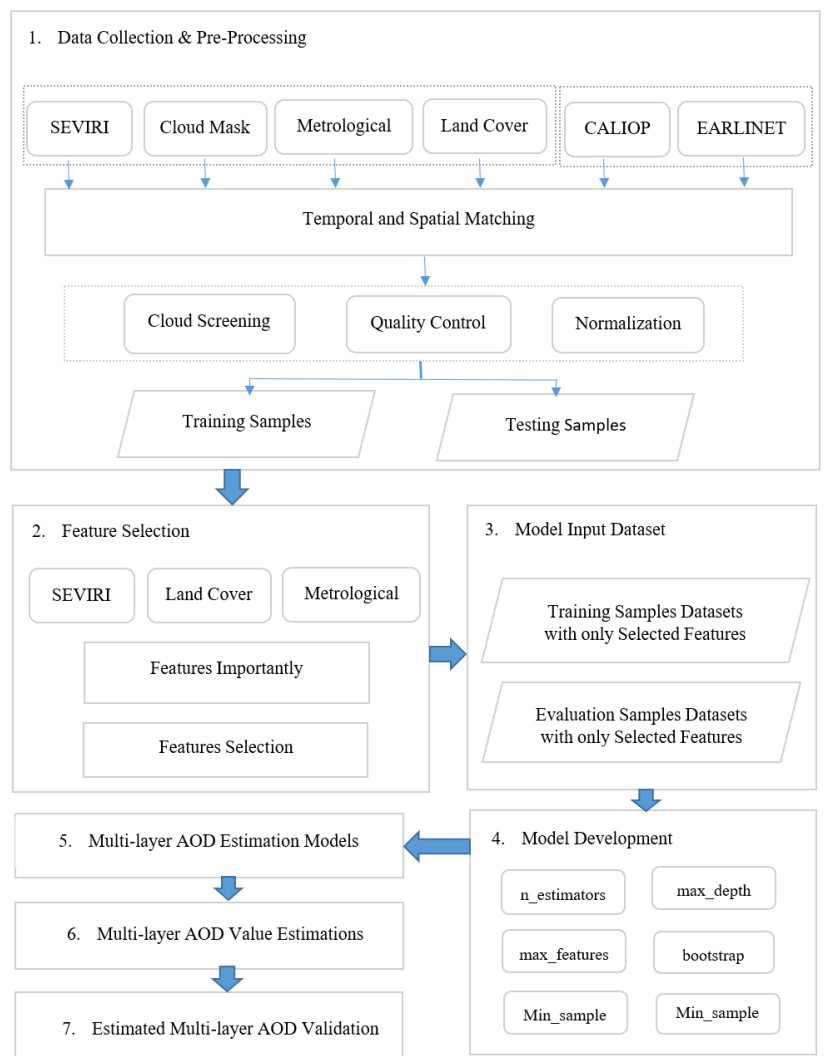

**Figure 4. Research framework for developing machine learning models to estimate SEVIRI multi-layer AOD values.**

## 3.1 Pre-Processing

To ensure a robust model for estimating AOD values at suitable 3D resolutions, this study integrates data from various sources, including satellites and ground-based observations. To address spatial-temporal sampling disparities, we employ a co-location approach where data from multiple sources, such as satellites and ground-based observations, are matched within a ±30 minutes timeframe and within a 3 km radius of the study area (Kittaka et al., 2011; Redemann et al., 2012; Han et al., 2017; Liu et al., 2018a). This method harmonizes disparate datasets, enhancing the reliability and comprehensiveness of our analysis. The subsequent preprocessing stages necessary for data refinement and analysis are elaborated upon in the following subsections.

### 3.1.1 SEVIRI

Utilizing SEVIRI data necessitates a critical preprocessing step involving co-referencing and applying geometric corrections. The Data Tailor tool, accessible at https://www.eumetsat.int/data-tailor, serves as a valuable spatial resource introduced in recent years. It simplifies the definition of coordinate systems, image systems, cutting ranges, expected output types, and requisite file extensions for the output data. Estimating AOD values requires the conversion of radiance to reflectance for the SEVIRI reflective bands (VIS06, VIS08, and NIR16), and equivalent brightness temperature for the remaining eight bands. To achieve this, we computed the Bidirectional Reflectance Factor (BRF) for the SEVIRI warm channels using Equation (1) proposed by the European Organization for the Exploitation of Meteorological Satellites (2012):

$$r_{\lambda_i} = \frac{\pi \cdot R_{\lambda_i} \cdot d^2(t)}{I_{\lambda_i} \cdot \cos(\theta(t,x))}, \tag{1}$$

where $i$ denotes the channel number (1 = VIS06, 2 = VIS08, 3 = NIR16, 4 = HRV), $r_{\lambda_i}$ represents the Bidirectional Reflectance Factor (BRF) for channel $\lambda_i$, $R_{\lambda_i}$ stands for the measured radiance in mW·$m^{-2}$·$sr^{-1}$·$(cm^{-1})^{-1}$, d(t) signifies the Sun-Earth distance in Astronomical Unit (AU) at time $t$, $I_{\lambda_i}$ signifies the band solar irradiance for channel $\lambda_i$ at 1 AU in mW·$m^{-2}$·$sr^{-1}$·$(cm^{-1})^{-1}$, and $\theta(t,x)$ denotes the Solar Zenith Angle in radians at time $t$ and location $x$. The equivalent brightness temperature ($T_b$) of a satellite observation is defined as the temperature of a black body emitting the same amount of radiation. Therefore, the brightness temperature follows the form of Equation (2).

$$T_b = \frac{C_2 \, v_c}{\alpha \, \log C_1 \, v_c^3 / \bar{R} + 1} - \frac{\beta}{\alpha}, \tag{2}$$

Using the observed radiances $\bar{R}$ (in $mW m^{-2} sr^{-1} (cm^{-1})^{-1}$) and radiation constants $C_1 = 2hc^2$ and $C_2 = hc/k$, where $c$, $h$, and $k$ represent the speed of light, Planck's constant, and the Boltzmann constant respectively, the regression coefficients $v_c$, α, and β are determined through non-linear regression analysis. This analysis is conducted on a pre-calculated lookup table generated for the various SEVIRI channels, as delineated in Table 1 (Tjemkes et al., 2012).

To further enhance the preprocessing workflow, we applied the SEVIRI cloud mask product to the data. This product categorizes pixels with values 00, 11, and 22, corresponding to clear, partially cloudy, and cloudy conditions, respectively. To integrate the cloud mask into the data, we utilized the transformation:

$$\text{Adjusted bands} = |\text{Cloud mask value} - 2| \cdot B_i \tag{3}$$

This ensures that data from cloudy or partially cloudy pixels are appropriately weighted or excluded, preserving the accuracy of AOD retrieval for clear-sky conditions. By incorporating the cloud mask product, the preprocessing step effectively eliminates biases introduced by cloud contamination, ensuring the reliability of subsequent AOD estimations.

**Table 1.  Values for the regression parameters.**

| Channel No. | Channel ID | $v_c$ , $cm^{-1}$ | $\alpha$ | $\beta$, K |
|---|---|---|---|---|
| 4 | IR 3.9 | 2567.330 | 0.9956 | 3.410 |
| 5 | WV 6.2 | 1598.103 | 0.9962 | 2.218 |
| 6 | WV 7.3 | 1362.081 | 0.9991 | 0.478 |
| 7 | IR 8.7 | 1149.069 | 0.9996 | 0.179 |
| 8 | IR 9.7 | 1034.343 | 0.9999 | 0.060 |
| 9 | IR 10.8 | 930.647 | 0.9983 | 0.625 |
| 10 | IR 12.0 | 839.660 | 0.9988 | 0.397 |
| 11 | IR 13.4 | 752.387 | 0.9981 | 0.578 |

**3.1.2 CALIOP**

In this study, to mitigate the impact of cloud contamination and retrieval errors on CALIOP AOD retrieval, our screening methods closely follow the guidelines established by Winker et al., 2013. We employ various quality filters to identify and filter aerosol pixels, including CAD scores, extinction QC flags, and uncertainty values. Specifically, we utilize a CAD score range outside [-100, -20] to address uncertainties in cloud-aerosol discrimination, ensuring the selection of cloud-free pixels with high confidence. Additionally, we apply extinction quality control flags with values 0 and 1 to filter extinction retrievals with high confidence. This includes constrained retrievals utilizing transmittance measurements and unconstrained retrievals where the initial LiDAR ratio remains unchanged in iterations. Furthermore, we exclusively consider daytime profiles in this study. Uncertainty flags associated with extinction coefficients are employed for data screening. Range bins with an uncertainty flag value of 99.9 $km^{-1}$ are excluded from the analysis, following the methodology outlined by Winker et al., 2013.

**3.1.3 Land Cover Product**

Considering that the original MCD12Q1 product is stored in a HDF and utilizes the sinusoidal projection, several data pre-processing steps are required. These steps encompass format conversion, reprojection, resampling, image mosaicking, and

sub-area masking. To execute these tasks, we employ the pyModis Free and Open-Source Python-based library. This tool enables the conversion of MODIS HDF data format into Geotiff format and facilitates the conversion of data projection from SIN to WGS84/UTM. Additionally, it facilitates image mosaicking and sub setting. Moreover, to enable comparison between the MCD12Q1 and SEVIRI datasets, the spatial resolution of MCD12Q1 is resampled at 3 km using the nearest neighbor resampling method. This method preserves the gray values of the original image, unlike bilinear interpolation or cubic convolution interpolation methods, which may alter them.

## 3.2 Machine learning Models and parameter Tuning

In this study, our primary objective is to develop a machine learning model to estimate SEVIRI AOD values at various altitudes—1.5 km, 3 km, 5 km, and 10 km—using CALIOP's vertical profiles across the European continent. We employ two distinct machine learning algorithms, RF and XGB, to train layering models. Both RF and XGB adopt an ensemble approach, which involves constructing and aggregating multiple decision trees (Breiman, 2001; Chen and Guestrin, 2016). In RF, each tree is built using a bootstrap sample of the data, with nodes determined by the best subset of randomly selected predictors (Breiman, 2001). These trees are then averaged to obtain a final ensemble prediction. Conversely, XGB implements the gradient boosting method, where trees are interdependent as newly trained trees are constructed based on previous trees, incorporating their ability to predict the residuals of prior trees (Chen and Guestrin, 2016). In both RF and XGB, all trained trees are combined to make the final prediction.

We systematically explored various parameter combinations for each machine learning model. Parameters such as the number of decision trees (N_estimators), the number of variables considered for splitting at each node (max_features), and the maximum depth of each decision tree (max_depth) for RF, as well as parameters including the number of gradient boosting rounds or decision trees (n_estimators), minimum sum of instance weight (Min_sample split), maximum depth of each decision tree (max_depth), and minimum number of samples required to be at a leaf node (Min_sample leaf) for XGB, were optimized using a grid search algorithm. This algorithm exhaustively searches through a specified subset of the hyperparameter space. We set up a grid of possible values for each hyperparameter to be tuned, as illustrated in the "Specific Search Range" column in Table 2. For each combination of hyperparameters in the grid, the algorithm trains the model using the training data and evaluates its performance through cross-validation. The performance of each hyperparameter combination is measured using several specified evaluation metrics. Finally, the combination of hyperparameters that results in the best performance on the validation set is selected, as shown in the "Optimum Value" column in Table 2. This optimal set of parameters is then used to train the final model on the entire training dataset. For a comprehensive overview of the optimized parameters, refer to Table 2.

**Table 2. The control parameter for tuning the machine learning models.**

| Model | Parameter | Specific search range | Optimum value |
|---|---|---|---|

| | | | |
|---|---|---|---|
| | n_estimators | 50 to 150 | 150 |
| | max_features | [auto, sqrt, log2] | sqrt |
| RF | max_depth | [5,10,20] | 20 |
| | bootstrap | [True, False] | False |
| | n_estimators | 50 to 500 | 100 |
| | max_depth | [5,10,20] | 20 |
| XGB | Min_sample split | 0.1 to 1 | 0.3 |
| | Min_sample leaf | 3 to 10 | 8 |

## 3.3 Model Training and Evaluation

Data partitioning is essential for training and evaluating machine learning models, especially when working with time-series data where temporal autocorrelation might bias model performance. In this study, the dataset was partitioned by year to ensure temporal independence between training and testing data, addressing potential autocorrelation issues and enabling robust

model evaluation. Specifically, the data from 2017 and 2018 were used for model training, while the 2019 data were reserved exclusively for testing. This approach was chosen following an analysis of the feature distributions of SEVIRI bands ($B_1$ to $B_{11}$), P, T, LC, Ws, Wd, and multi-layer AOD values ($AOD_{1.5}$, $AOD_3$, $AOD_5$, and $AOD_{10}$) over the different years, as shown in supplementary Fig. S1. The distribution represented in Fig. S1 reveal consistent patterns in patterns between 2017–2018 and 2019, with minimal variation in their shapes. This similarity confirms that the temporal separation does not introduce

significant distributional shifts that might impact model generalization. In other words, the model's performance on the 2019 data would provide an unbiased evaluation of its predictive ability. This separation minimizes temporal autocorrelation, ensuring robust and unbiased model assessment.

During the training phase of our machine learning models, we leveraged datasets spanning diverse temporal periods and geographical regions where both SEVIRI and CALIOP data were accessible. However, following this training phase, the

algorithms function autonomously, relying solely on SEVIRI data as their input. This advancement enables us to estimate AOD values at four specified vertical layers within each pixel of the SEVIRI dataset, based on a single SEVIRI observation along with its associated meteorological data and land cover data, covering the entire study area.

Evaluation of the multi-layer AOD estimating models involved statistical metrics such as the coefficient of determination ($R^2$), Pearson correlation coefficient (R), root mean square error (RMSE), and mean absolute error (MAE). The selection of the

optimal model was based on higher $R^2$ and R values, along with lower RMSE and MAE scores. Additionally, we conducted a validation analysis of estimated multi-layer AOD values with EARLINET AOD profiles on a continental scale to ascertain the model's performance.

## 4 Results and Discussion

In this paper, our primary aim is to develop a machine learning model capable of retrieving AOD across four distinct vertical layers: 1.5, 3, 5, and 10 km. To accomplish this, we utilized two well-established machine learning models, XGB and RF, previously employed in related studies. These models were trained on SEVIRI data spanning the European continent from 2017 to 2019. Our objective was to estimate sub-hourly AOD values, approximately every 15 minutes, at a spatial resolution of 3 km × 3 km.

To explore the relationship between AOD and potential predictor variables, we conducted a correlation analysis experiment utilizing the Pearson Correlation Coefficient (PCC, Benesty et al., 2009). Furthermore, we evaluated the influence of land cover and meteorological data as input variables for the machine learning models in estimating multi-layer AOD values from SEVIRI data, with a specific focus on identifying the most optimal model. Moreover, we conducted training and testing of the machine learning models across various temporal scales, including annual and seasonal analyses. Subsequently, we assessed the performance of each model using independent satellite and ground-based AOD profiles, employing evaluation metrics such as R², R, MAE, and RMSE. Finally, multi-layer AOD values for two aerosol events—Saharan dust from March 13 to March 18, 2022, and a volcanic eruption on August 14, 2023—are presented as maps to evaluate the model's ability to detect and characterize distinct aerosol signatures across these four layers. In the subsequent sections, we will provide a comprehensive review of the results derived from the aforementioned assessments.

### 4.1 Validation of Estimated AOD with Satellite Retrieval AOD

#### 4.1.1 Feature Importance

According to established radiative transfer theory (Tsang et al., 1984; Zege et al., 1991), the spectral signal captured by a satellite sensor at the TOA is intricately shaped by various factors, including the composition, size distribution, and altitude of aerosols, as well as atmospheric molecules such as water vapor. These factors have a direct impact on the retrieval of AOD values. Consequently, SEVIRI reflectance and brightness temperature across Bands 1 to 11 ($B_1$ to $B_{11}$) were identified as critical features for this analysis. The relationship between AOD and all candidate features—including spatial features such as latitude (lat) and longitude (lon), temporal features including year, month, and day, as well as meteorological data like P, T, Ws, and Wd, and LC—was investigated through a correlation analysis. This analysis, illustrated in Fig. 5, utilized the PCC as the chosen filtering method. The findings underscored that the majority of selected features in this study exhibited significance levels exceeding 1%.

As illustrated the PCC results (Fig. 5), aerosol dynamics in the 0–1.5 km layer of the study area are strongly influenced by geographic and temporal factors, characteristic of surface-dominated conditions. High correlations of AOD with lon (23.92%) and lat (22.41%) highlight the impact of location-specific emissions and regional transport patterns, while significant correlations with day (11.59%) and month (6.19%) indicate the role of diurnal and seasonal cycles driven by emissions, meteorological changes, and boundary layer dynamics. Thermal infrared sensitivity also plays a role, as indicated by the strong

correlations with $B_7$ (3.73%) and $B_8$ (3.07%), the highest among SEVIRI bands. This suggests a potential link between surface temperature variations and AOD in the lower atmosphere. Moreover, the correlation with $B_1$ (3.06%) points to aerosols interacting with visible light, likely from urban/industrial emissions, biomass burning, and potentially dust. Meanwhile, meteorological factors such as Ws (1.31%), Wd (1.22%), T (1.16%), and P (1.1%) are relatively weak compared to geographic and temporal factors. Finally, LC (0.54%) shows the weakest correlation, suggesting a limited direct influence on AOD, potentially masked by stronger influences from emissions, transport, and meteorological factors.

In the 1.5–3 km layer of area of interest, aerosol dynamics remain strongly correlated with lon (23.15%) and lat (21.39%), though slightly decrease compared to the 0-1.5 km layer. This suggests a transition towards atmospheric stability and a more synoptic-scale patterns. Day (11.29%) and month (5.86%) still exhibit notable correlations, reflecting the ongoing impact of diurnal and seasonal cycles on AOD, albeit slightly weaker than in the lower layer. The visible band $B_1$ (3.86%) shows the strongest correlation among SEVIRI bands, potentially reflecting a change in aerosol composition or properties compared to the lower layer. This could be due to an increased influence of transported aerosols, potentially with different spectral characteristics. $B_7$ (3.48%) remains strongly correlated, suggesting continued sensitivity to thermal characteristics. Enhanced transport influence is evident, as Wd (1.34%) shows a stronger correlation than Ws (1.29%), emphasizing the role of transport pathways in AOD distribution at this altitude. Additionally, the correlation with P (1.3%) points to an emerging link between atmospheric stability and aerosol accumulation in this layer. This aligns with other studies' discussion on PBLH and its role in controlling pollutant dispersion, as PBLH is influenced by factors like temperature and stability. LC correlation slightly increases to 0.63%, still remaining relatively weak overall.

In the 3–5 km layer of research area, correlations with lon (19.42%) and lat (17.32%) decrease further, indicating a reduced influence of local sources and an increased dominance of long-range transport and large-scale atmospheric circulation patterns. Correlations with day (9.87%) and month (6.1%) also weaken, suggesting a shift from surface-driven cycles to broader atmospheric processes. The high correlation with $B_1$ (5.38%) points to a prominent presence of aerosols interacting with visible light. Strong correlations with $B_7$ (4.73%) indicate effect of thermal characteristics on AOD at this level. Wd (1.18%), Ws (1.1%), T (1.09%), and P (0.99%) correlations remain at similar levels as the previous layer, suggesting continued influence of transport and stability.

In the 5–10 km layer of target region, free tropospheric dynamics dominate aerosol behaviour, geographic correlations with lon (22.92%) and lat (21.46%) show a slight increase compared to the 3-5 km layer, potentially indicating the influence of large-scale transport patterns in the free troposphere. Day (8.59%) and month (4.91%) correlations further weaken, reinforcing the diminishing impact of surface-driven cycles. $B_1$ (5.25) still showing the highest correlation, indicating aerosols that interact with visible light and are likely transported from lower altitudes or distant sources. $B_7$ (4.6%) also maintains a strong correlation, indicating a potential sensitivity to upper-level atmospheric dynamics. Notably, P (1.41%) now shows the strongest correlation among meteorological variables, surpassing Wd (1.13%), T (1.1%), and Wd (0.84%). This emphasizes the critical role of atmospheric stability in controlling AOD distribution in the free troposphere. Despite meteorological data showing low statistical importance (below 2%) in estimating AOD across all altitude layers, they have significant physical relevance in

processes unique to each layer. LC (0.42%) shows the weakest correlation across all layers, further suggesting its limited direct influence on AOD at this altitude.

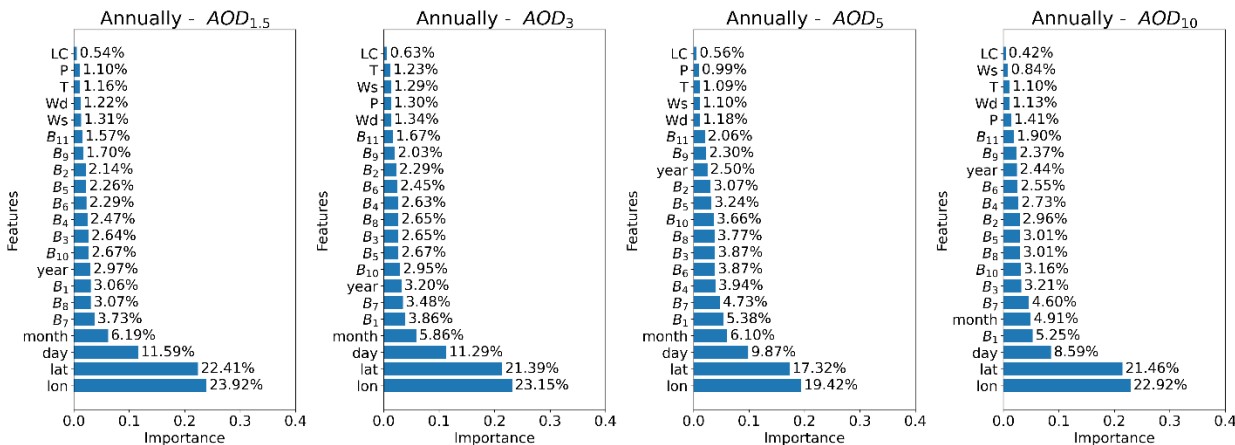

**Figure 5. The Importance of input features in the retrieval of SEVIRI multi-layer AOD values, as determined by the PCC.**

### 4.1.2 Meteorological data and Land cover Feature selection

As previously noted, LC, T, P, Ws, and Wd are key features in AOD estimation. To further understand their impact on machine learning model performance in SEVIRI multi-layer AOD values, we conducted 16 cases of experiments with varied meteorological and LC feature combinations, validated using CALIOP AOD retrievals. Our analysis, depicted in Fig. 6 and supplementary Tables S1-S2, is summarized using statistical metrics like $R^2$, R, RMSE, and MAE.

Our findings indicate that, for most cases across annual and seasonal datasets, combining these features with $B_i$ has negligible impact on the 1.5 km layer. In contrast, integrates Ws and Wd with SEVIRI bands (Case 5), consistently yields the highest $R^2$ and lowest RMSE values across both annual and seasonal datasets. Incorporating these features significantly enhances $R^2$ values across models, with substantial increases ranging from 0.75 to 0.99 and 0.89 to 0.99 observed in $R^2$, and decreases ranging from 0.075 to 0.009 and 0.033 to 0.002 in RMSE, for both XGB and RF models from Case 1 to Case 5 in the 10 km layer. These statistical values underscore the crucial role of Ws and Wd in influencing the spatial and temporal properties of atmospheric aerosols, particularly in the 10 km layer. Physically, Ws and Wd are known to be primary drivers of aerosol transport across various spatial scales. High Ws can lift dust and other particulate matter into the atmosphere, while Wd affects the regional and long-range advection of aerosols. This mechanism is particularly impactful in the 5-10 km altitude range, where aerosols experience less drags and can be transported over long distances with minimal settling, especially in dry conditions (Pérez et al., 2006a, b; Georgoulias et al., 2016; Nicolae et al., 2019).

However, integrating T and P features, as seen in cases 3, 4, and 9, notably enhances AOD accuracy at 3, 5, and 10 km altitudes. This improvement is attributed to P reflecting changes in aerosol vertical layers, influencing aerosol diffusion capacity, while T is closely linked to atmospheric aerosol distribution by altering air movement dynamics. T affects atmospheric stability by

controlling the thermal stratification of air masses; higher T can destabilize the atmosphere, promoting vertical mixing and lifting of aerosols. However, the impact of T on aerosol concentration diminishes with altitude, especially in stable layers where thermal inversion limits upward transport (Pérez et al., 2006a, b; Choobari et al., 2014; Xu et al., 2024). P, on the other hand, can reflect shifts in the PBL height, which influences the vertical distribution of aerosols. A higher PBL allows more

aerosols to disperse vertically, enhancing their presence in layers such as $AOD_5$, while a lower PBL restricts aerosols closer to the surface. Despite these influences, T and P are generally less effective than wind-related factors (cases 6, 7, and 8), particularly in upper layers, because wind-driven advection predominantly controls the lateral movement of aerosols, especially during seasonal changes when wind dynamics vary substantially (Nicolae et al., 2019; Georgoulias et al., 2016; Ortiz-Amezcua et al., 2017; Granados-Muñoz et al., 2016). However, T and P impact PBL dynamics, but these effects may be

limited in certain seasons due to more stable atmospheric conditions, where lower T and P fluctuations are less conducive to vertical mixing. For example, case 12 including T and P may show less impact on $AOD_5$ and $AOD_{10}$ layers in winter, when aerosols tend to stay close to the surface due to limited vertical convection. In contrast, warmer seasons promote vertical convection, enhancing the influence of T and P in predicting AOD values across altitude layers. Additionally, the proposed learning models can capture complex, nonlinear relationships among features, but it may not always prioritize individual

variables unless they strongly affect the target variable. As a result, even if T or P is expected to influence AOD, their impact may be overshadowed by more influential Ws and Wd (cases 13, 14, 15, and 16).

Conversely, LC plays a more localized role in AOD estimation, particularly in lower layers ($AOD_{1.5}$), as it impacts the sources and types of aerosols present near the surface. However, the effect of LC diminishes with altitude due to decreased influence on vertical transport; aerosols released from surface sources are progressively diluted as they disperse upwards. This

observation is confirmed by our findings that cases with LC (e.g., Cases 2, 8, 13, 14, and 16) did not consistently outperform those with purely meteorological features, especially in higher layers ($AOD_5$ and $AOD_{10}$). Physically, this limitation arises from the fact that the vertical distribution of aerosols across different atmospheric layers over Europa troposphere is more heavily influenced by continental and regional transport patterns, atmospheric stability, and meteorological conditions than localized land cover characteristics (Zhao et al., 2019).

In conclusion, the validation of our models using CALIOP AOD retrievals highlights the robustness of our findings, particularly the critical role of Ws and Wd in enhancing AOD estimation accuracy. The consistency of these findings across the RF and XGB models, evaluated at different temporal scales (annual and seasonal), highlights the critical role of Ws and Wd in AOD estimation at both the 5 km and 10 km layers. Consequently, we prioritize Ws and Wd, along with $B_i$, as the preferred input features for our models due to their demonstrated impact on improving AOD estimation accuracy.

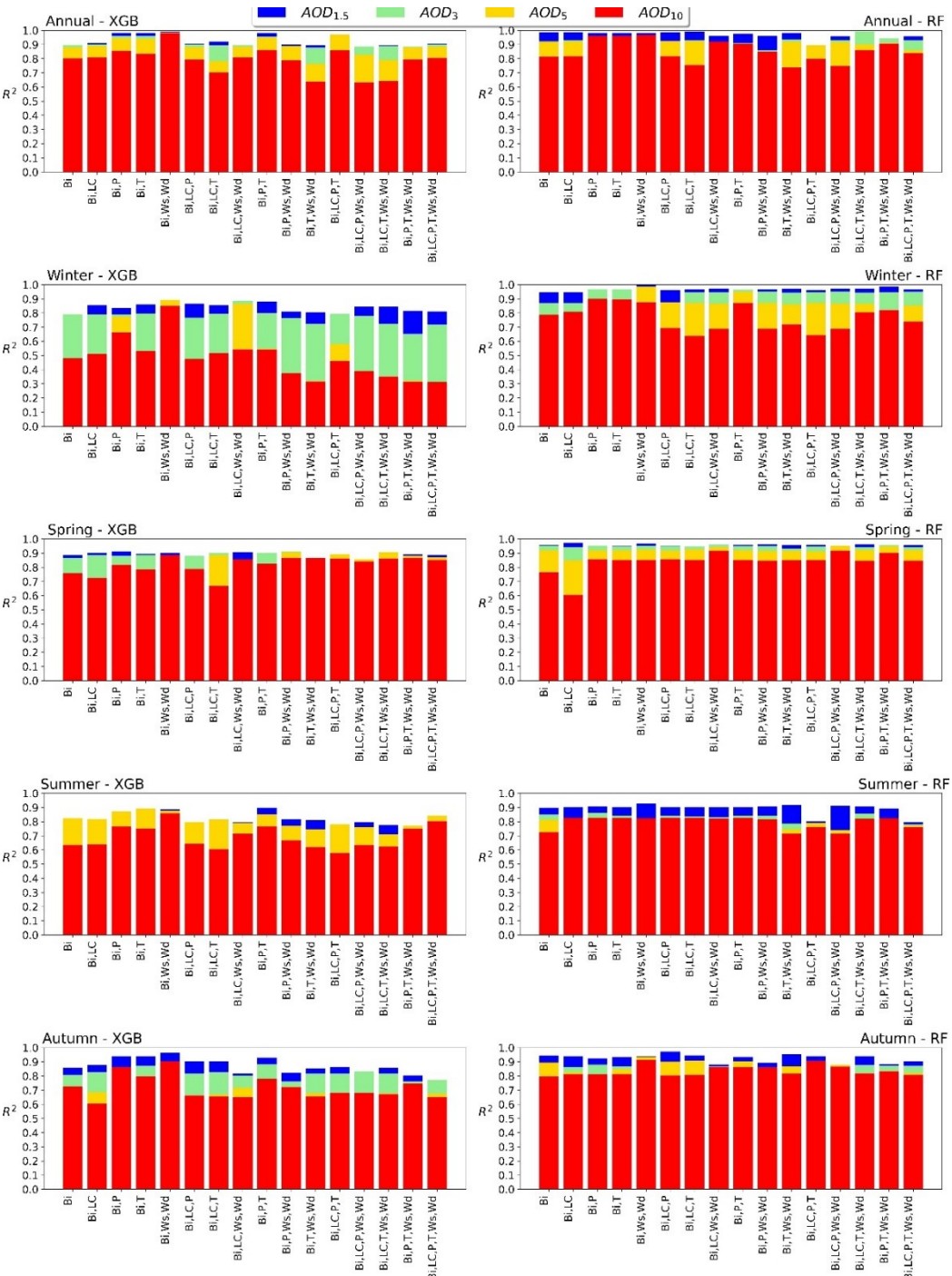

**Figure 6. The impacts of input features on the retrieval of SEVIRI multi-layer AOD values, represented as $R^2$ metrics, for the RF and XGB models. Each row displays the results for the annual period and the four seasons (Winter, Spring, Summer, and Autumn). The four colors in each bar plot indicate the $R^2$ values for AOD at 1.5, 3, 5, and 10 km layers.**

### 4.1.3 Validation of Seasonal Modeling

Considering the substantial seasonal variations in aerosol distribution and meteorological conditions, we sought to evaluate whether adapting our proposed modelling approach to specific seasons could improve the accuracy of multi-layer AOD retrievals. Following the methodology outlined in Section 3, we partitioned the sample dataset, derived from 2017 to 2019 data, into four segments based on seasonal distinctions: Winter (January, February, and March), Spring (April, May, and June), Summer (July, August, and September), and Autumn (October, November, and December), as detailed in Table 3. Subsequently, we trained individual machine learning models on these seasonal and annual datasets spanning 2017 to 2018. For this analysis, we separately estimated SEVIRI multi-layer AOD values for the year 2019 using both the seasonal and annual models. Detailed seasonal validation findings, including $R^2$ and RMSE metrics, are delineated as highlighted values in Table S1 and S2.

**Table 3. Number of Samples Used to machine learning models train in this Study.**

| Data | Period | All | Winter | Spring | Summer | Autumn |
|------|--------|-----|--------|--------|--------|--------|
| Train | 2017-2018 | 37325 | 6830 | 9038 | 12108 | 9349 |
| Test | 2019 | 18117 | 2548 | 4703 | 7560 | 3306 |

The XGB model exhibited acceptable performance across different seasons, with $R^2$ (RMSE) values for the 1.5 km layer as follows: 0.901 (0.0103) for spring, 0.889 (0.0347) for summer, 0.966 (0.0265) for autumn, and 0.881 (0.0392) for winter. In comparison, the RF model demonstrated improvement, boasting $R^2$ values of 0.97, 0.927, 0.937, and 0.999, with corresponding RMSE values of 0.0211, 0.0278, 0.052, and 0.0023 for spring, summer, autumn, and winter, respectively. Similarly, both the XGB and RF models demonstrated satisfactory performance across other layers, with $R^2$ ranging from 0.81 to 0.95, 0.80 to 0.98, and 0.82 to 0.91 for the 3 km, 5 km, and 10 km layers, respectively. Performances of models generally tend to decrease in the upper layers compared to the lower layers. Due to the prevalent types and sizes of existing aerosols throughout most of the year, with aerosol distribution in Europe predominantly concentrated within the 1.5 and 3 km atmospheric layers. Consequently, $R^2$ and R metrics demonstrate higher values in these layers compared to the 5 and 10 km layers. Conversely, RMSE and MAE metrics are elevated at the 1.5 and 3 km layers but lower at the 5 and 10 km layers. This pattern arises from the typically higher aerosol concentrations occurring in the lower atmospheric layers, juxtaposed with lower AOD values observed in the 5 and 10 km layers.

However, the XGB Annually model exhibited strong performance, achieving the RMSE values (0.0091, 0.0134, 0.0066, and 0.0059) and the $R^2$ values (0.993, 0.974, 0.985, and 0.981). Similarly, the RF Annually model produced notable results, with $R^2$ values of 0.98, 0.962, 0.939, and 0.968, along with RMSE values of 0.015, 0.010, 0.0112, and 0.0066, respectively. In conclusion, annual models demonstrate consistently high predictive accuracy, with XGB showing slightly stronger performance in terms of both $R^2$ and RMSE. The minimal RMSE and high $R^2$ values indicate that both models effectively capture the multi-layer AOD values across the altitude layers on an annual basis. Compared to seasonal models, the annual

models offer greater stability and reduced variability, as they are less affected by seasonal meteorological changes that can alter aerosol distribution, such as summer convection and winter temperature inversions. This consistency in annual performance provides a more reliable basis for long-term AOD retrieval, making annual models preferable for retrieving multi-layer AOD values for SEVIRI data. Therefore, we considered the annual models as the desired models for retrieving multi-layer AOD values of SEVIRI.

### 4.1.4 Comparison of the models

Figure 7 presents scatterplots illustrating multi-layer AOD values estimated using the proposed annual XGB (Fig. 7a-d) and RF (Fig. 7e-h) models, compared with CALIOP-retrieved AOD profiles over Europe troposphere in 2019. Each subplot includes the number of points and mentioned metrics i.e. $R^2$, R, RMSE, MAE, Bias, and linear regression equations to facilitate clear and thorough analysis.

Both models exhibit a strong correlation between the estimated values and retrievals. However, the XGB model demonstrates slightly superior performance, with $R^2$ (R) values of 0.993, 0.974, 0.985, and 0.981 (0.997, 0.989, 0.993, and 0.991) for the 1.5, 3, 5, and 10 km layers, respectively. In comparison, the RF model shows lower $R^2$ (R) values of 0.980, 0.962, 0.939, and 0.968 (0.993, 0.984, 0.972, and 0.985) for the same layers. The XGB model consistently outperforms in estimating AOD across all layers. However, the RF model demonstrates relatively higher accuracy in the 1.5 km layer compared to other layers, highlighting a significant performance gap when compared to XGB. Overall, the minimal variation in $R^2$, R, RMSE, and MAE across the models suggests comparable estimation capabilities. However, a detailed analysis reveals the superior accuracy of the XGB model in capturing AOD values, as evidenced by the slope values in Fig. 7a-d. In contrast, the slope values in Fig. 7e-h indicate that the RF model tends to slightly underestimate AOD values. In summary, while both models demonstrate proficiency, the XGB model outperforms the RF model, particularly in its accuracy for higher altitude layers, thereby providing more reliable AOD estimations.

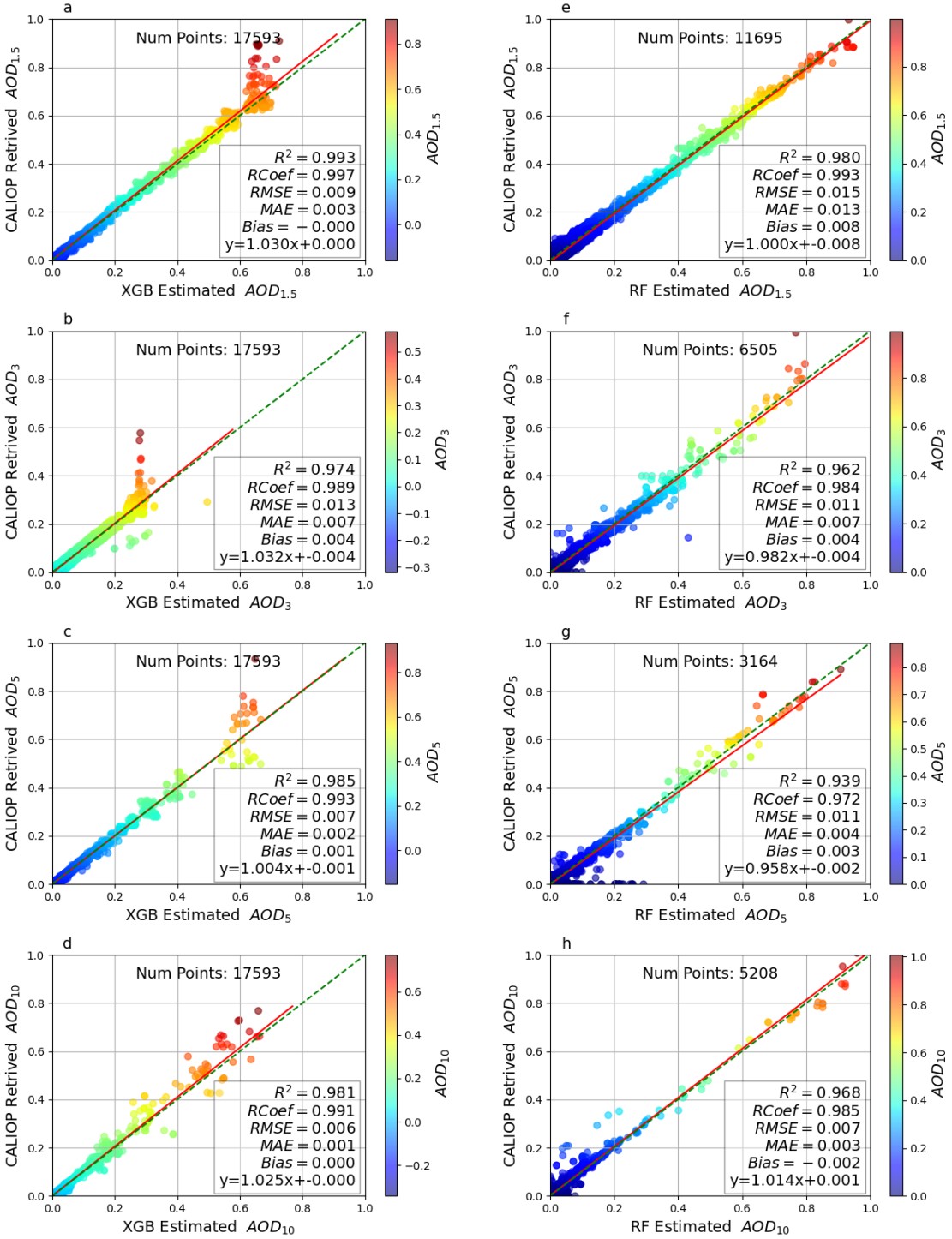

**Figure 7. Scatterplots comparing the estimated SEVIRI multi-layer AOD values derived from the proposed machine learning models with the CALIOP AOD profiles for the year 2019. The red line represents the linear fit between the two datasets.**

## 4.2 Validation of Estimated AOD with Ground LiDAR Retrievals

To further validate our top-performing models, annual XGB and RF, we conducted an extensive analysis using data from eight EARLINET stations across Europe in 2020. We analysed pixels within a 3 km radius of each station to compare and validate SEVIRI multi-layer AOD estimates against EARLINET values. Figure 8 presents the comparison using linear regression and validation metrics, with scatterplots for AOD at four layers. The XGB model demonstrates stronger agreement with EARLINET profiles, achieving $R^2$ values of 0.86, 0.79, 0.75, and 0.59, and RMSE values of 0.02, 0.01, 0.01, and 0.005, respectively. In contrast, the RF model shows weaker correlations, with $R^2$ values of 0.83, 0.26, 0.52, and 0.16, and RMSE values of 0.024, 0.022, 0.021, and 0.007. This can be attributed to XGB's ensemble nature and its ability to reduce bias through boosting, enabling it to handle complex and diverse datasets more effectively (Ahmed et al., 2023).

When comparing the $R^2$ metrics of XGB AODs across different layers, it was found that XGB AODs exhibited lower $R^2$ values with EARLINET at the 10 km layer but showed significant improvement at the 1.5, 3, and 5 km layers, with $R^2$ values of 0.86, 0.79, and 0.75, respectively. This indicates a stronger correlation between XGB AOD estimations and EARLINET retrievals in these layers compared to the 10 km layer, which had an $R^2$ value of 0.593. This trend is consistent with other evaluation metrics. Closely scrutinizing Fig. 8, it becomes apparent that specific points revealing notable discrepancies between EARLINET and XGB AOD values. To determine the root cause of these outliers, the data were color-coded based on AOD values, revealing that the majority of outliers occurred when EARLINET retrieved low AOD values in each layer. At these points, the XGB model tends to overestimate. This tendency contributed to a low $R^2$ value (0.593) in the linear regression for the 10 km layer, as this layer contains small AOD values (0-0.05). Furthermore, the slope of the regression line for the XGB model ranges from 0.82 to 0.99, indicating that for every unit increase in the estimated values, the corresponding EARLINET AOD values increase by slightly less than one unit. This suggests a tendency for the XGB model to overestimate EARLINET AOD across all layers.

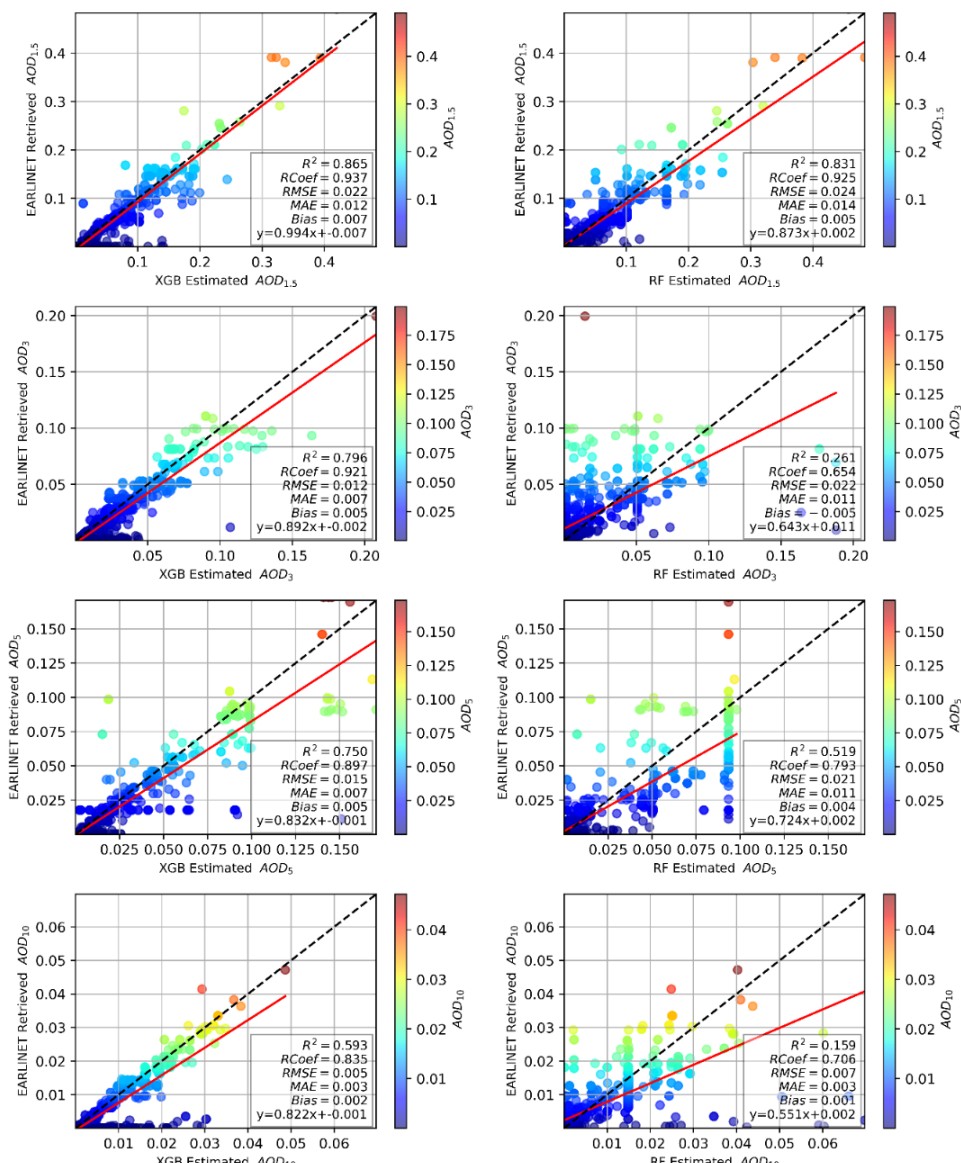

**Figure 8. Scatterplots comparing the estimated SEVIRI multi-layer AOD values derived from the proposed machine learning models with the EARLINET AOD profiles across 8 specified stations in Europe for the year 2020. The red line represents the linear fit between the two datasets. Note that the scales of the subplots vary due to the different ranges of AOD values at the various vertical layers (1.5, 3, 5, and 10 km).**

The statistical analysis of EARLINET AOD profiles at the eight specified stations, alongside estimated AOD values, is comprehensively presented in Table 4. The number of analyzed pairs (N) varies across stations, ranging from 12 at HPB to 387 at ATZ, providing robust validation of estimated AOD values with EARLINET AOD profiles. Metrics such as RMSE,

MAE, and Bias offer valuable insights into model performance at each station. A detailed analysis of the data reveals that the values of these metrics across the four layers are not consistently identical.

Performance varies significantly across stations and layers, with notable discrepancies observed at ATZ (Greece), particularly at the 1.5 km layer, which exhibits the highest RMSE of $3.1 \times 10^{-2}$. The substantial Bias at this station indicates that the model tends to consistently overestimate the AOD at this altitude. Conversely, performance improves at 3 km and 5 km, with RMSE values of $1.5 \times 10^{-2}$ and $1.8 \times 10^{-2}$, respectively. At this station, the model demonstrates the best performance at the 10 km layer, with an RMSE of $0.6 \times 10^{-2}$, compared to the other layers. These variations are likely due to frequent forest fires in Greece, as most smoke from these fires remains in the lower layers of the atmosphere (Nicolae et al., 2019). In contrast, the XGB model generally performs well at the SAL, HPB, and LLE stations, where both RMSE and Bias are minimal. The RMSE at the 1.5 km layer, ranging from $0.4 \times 10^{-2}$ to $2.2 \times 10^{-2}$ at the IPR, WAW, INO, and THE stations, alongside the low RMSE and Bias values across other layers, demonstrates good overall model performance at these stations. A closer examination reveals that RMSE and Bias metrics are often elevated at the 1.5 km and 3 km layers but lower at the 5 km and 10 km layers. This pattern arises from the typically higher aerosol concentrations in the lower atmospheric layers, compared with lower AOD values retrieved in the 5 km and 10 km layers.

The discrepancies between the estimated and retrieved values could stem from the different measurement techniques employed by satellite and ground-based systems. EARLINET utilizes ground-based LiDAR systems to capture backscattered light from aerosols within the atmosphere by looking upward, whereas satellite measurements are performed from above, looking down. In this configuration, the lower atmospheric layers attenuate the LiDAR signal, resulting in reduced power to penetrate the upper layers. This attenuation can complicate the detection of aerosols in the upper layers (Grigas et al., 2015; Nicolae et al., 2019). Furthermore, these limitations may be attributed to the constraints associated with the utilization of CALIOP AODs, particularly their reduced precision in low aerosol concentration scenarios. This reduced precision arises from the low signal-to-noise ratio under clean weather conditions, which is often insufficient to accurately detect weak aerosol layers on the aerosol extinction vertical profile. Because both transmitted and scattered light must traverse this portion of the atmosphere, highly diffuse and/or tenuous scattering aerosol layers below the CALIOP detection threshold are ignored in CALIOP's estimates of column AOD. Consequently, weak aerosol layers that are not detected would not be retrieved, leading to decreased retrieved AODs under clean weather conditions (Liu et al., 2018a, b).

Finally, the efficacy of the XGB model is clearly demonstrated by its ability to reliably estimate multi-layer AOD values compared to EARLINET retrieved AOD profiles across various European regions.

**Table 4. Station-based statistical analysis of XGB-estimated SEVIRI vs. retrieved EARLINET AOD values. All metric values are scaled by $\times 10^{-2}$.**

| ID | location | N | Layer | MAE | RMSE | Bias | ID | location | N | Layer | MAE | RMSE | Bias |
|---|---|---|---|---|---|---|---|---|---|---|---|---|---|
| INO | Romania | 13 | $AOD_{1.5}$ | 1.1 | 1.4 | 0.7 | THE | Greece | 50 | $AOD_{1.5}$ | 1.1 | 1.7 | 0.9 |
| | | | $AOD_3$ | 0.8 | 1.5 | -0.3 | | | | $AOD_3$ | 0.8 | 1.3 | 0.4 |
| | | | $AOD_5$ | 1.01 | 2.2 | 0.8 | | | | $AOD_5$ | 0.5 | 0.7 | 0.4 |
| | | | $AOD_{10}$ | 0.25 | 0.4 | 0.05 | | | | $AOD_{10}$ | 0.3 | 0.5 | 0.3 |
| IPR | Italy | 257 | $AOD_{1.5}$ | 1.5 | 2.2 | 1.0 | WAW | Poland | 28 | $AOD_{1.5}$ | 0.3 | 0.4 | 0.3 |
| | | | $AOD_3$ | 0.8 | 1.1 | 0.6 | | | | $AOD_3$ | 0.06 | 0.1 | 0.06 |
| | | | $AOD_5$ | 0.9 | 1.6 | 0.5 | | | | $AOD_5$ | 0.05 | 0.07 | 0.05 |
| | | | $AOD_{10}$ | 0.3 | 0.5 | 0.2 | | | | $AOD_{10}$ | 0.05 | 0.05 | 0.05 |
| ATZ | Greece | 387 | $AOD_{1.5}$ | 2.1 | 3.1 | 1.4 | HPB | Germany | 12 | $AOD_{1.5}$ | 0.07 | 0.1 | 0.07 |
| | | | $AOD_3$ | 1 | 1.5 | 0.8 | | | | $AOD_3$ | 0.04 | 0.05 | 0.04 |
| | | | $AOD_5$ | 1.03 | 1.8 | 0.7 | | | | $AOD_5$ | 0.01 | 0.02 | 0.02 |
| | | | $AOD_{10}$ | 0.3 | 0.6 | 0.3 | | | | $AOD_{10}$ | 0.02 | 0.03 | 0.02 |
| SAL | Italy | 13 | $AOD_{1.5}$ | 0.05 | 0.05 | 0.05 | LLE | France | 39 | $AOD_{1.5}$ | 0.2 | 0.26 | 0.16 |
| | | | $AOD_3$ | 0.04 | 0.05 | 0.03 | | | | $AOD_3$ | 0.08 | 0.1 | 0.08 |
| | | | $AOD_5$ | 0.008 | 0.01 | 0.008 | | | | $AOD_5$ | 0.03 | 0.03 | 0.02 |
| | | | $AOD_{10}$ | 0.04 | 0.04 | 0.04 | | | | $AOD_{10}$ | 0.03 | 0.03 | 0.02 |

## 4.3 Qualitative Validation

For qualitative validation of the estimated multi-layer AOD values, two distinct approaches were utilized. The first approach involved a visual comparison of the spatial trends of the estimated AOD values with CALIOP AOD retrievals for four specific days across various seasons in 2019: March 3 (11:57), April 30 (12:42), June 13 (10:57), and October 31 (12:27). The second approach concentrated on two noteworthy aerosol events: a significant dust intrusion that occurred over Europe from March

13 to March 18, 2022, and a volcanic eruption at Mount Etna in Italy on August 14, 2023. For these events, the estimation model generated multi-layer AOD values at 15-minute intervals. This high temporal resolution enabled detailed analysis of aerosol behaviour within each layer during these events.

Conducting the qualitative validation for an entire scene within the first approach is challenging due to the spatial and temporal resolution constraints of CALIOP. To address this limitation, SEVIRI scene pixels corresponding to CALIOP overpasses with temporal differences of less than four minutes were compared on the specified days. The results, illustrated in Fig. 9, indicate that the spatial trends of the estimations generally align well with the trends of CALIOP AOD retrievals in regions with both high and low AOD values across the four seasons. This alignment highlights the model's ability to provide reliable AOD estimates with enhanced temporal resolution, effectively complementing CALIOP AOD retrievals.

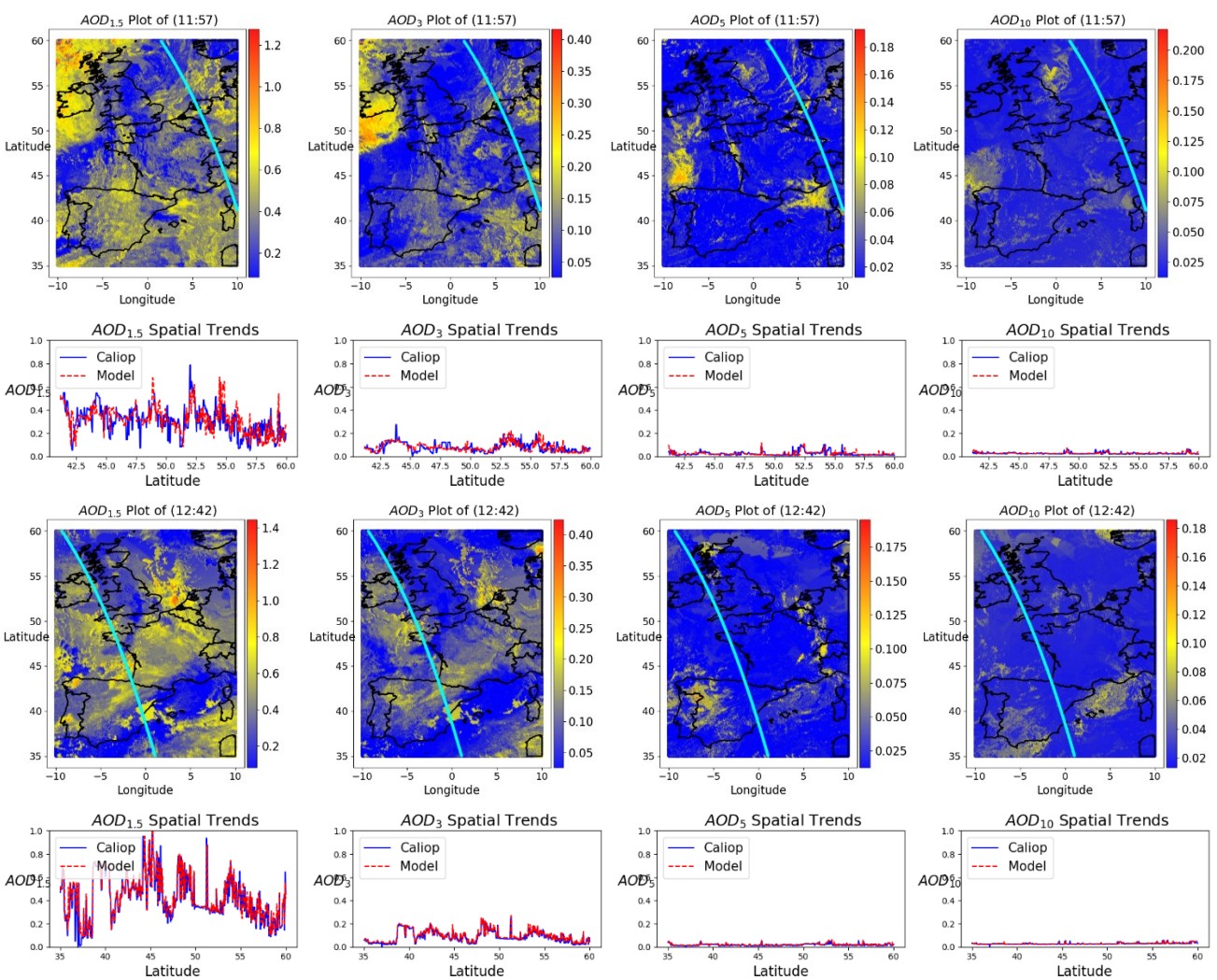

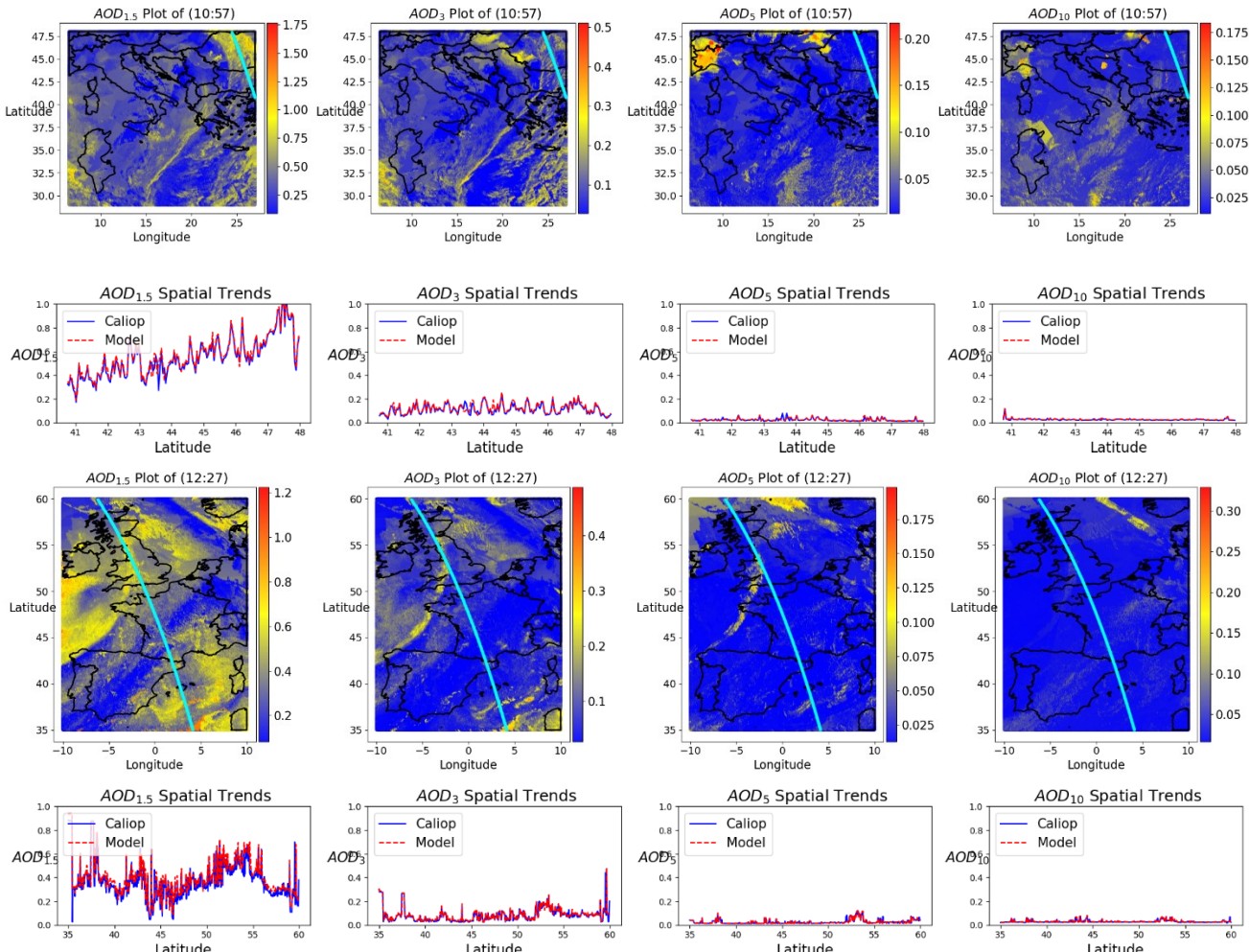

**Figure. 9.** Spatial distributions and trends of SEVIRI-retrieved multi-layer AOD values compared to CALIOP retrievals for four specific days in March 3 (11:57), April 30 (12:42), June 13 (10:57), and October 31 (12:27) across various seasons in 2019.

The subsequent approach for qualitative validation of the model on a regional scale involved analyzing two aerosol events. The first event examined was a substantial Saharan dust plume that traversed Western and Central Europe between March 13 and 18, 2022. SEVIRI scenes captured during this event, illustrated in Fig. 10, were visualized at specific hours to assess how well the aerosol vertical dispersion behavior aligned with the dynamic characteristics of the event.

5   On March 13, at 13:42 UTC, the event is characterized by concentrated aerosol presence at 1.5 km over southern Europe, particularly Spain, Italy, and the Mediterranean, with progressive dispersion at higher altitudes reaching central Europe. By March 14 at 14:42 UTC, the dust plume expands further into central and northern Europe at 1.5 km, while mid-altitudes ($AOD_3$ and $AOD_5$) show notable aerosol presence over the Iberian Peninsula and central Europe, reflecting the dust's horizontal

transport. At 10 km, concentrations remain low, indicating limited vertical penetration. Comparing these two days reveals a clear intensification and northward progression of the plume. On March 15, at 10:12 UTC, $AOD_{1.5}$ shows sustained high concentrations over southern Europe, with increased impact in southeastern Europe, including the Balkans. At mid-levels, the plume spreads further into central and eastern Europe, reaching countries like Germany and Poland, while at 10 km, faint concentrations persist over parts of southern and central Europe. By March 16 at 09:57 UTC, the plume shows signs of dissipation, with reduced intensity at 1.5 km across Spain, Greece, and the Mediterranean, and weaker signals at mid-altitudes over central Europe. On March 17 at 08:27 UTC, the dust plume is confined mainly to southern Europe at lower altitudes, while mid-altitudes exhibit limited spread and intensity, and 10 km shows negligible influence.

Finally, the spatial distributions on March 18 at 14:42 UTC, as shown in the figure, reveal a significant reduction in aerosol concentrations at all altitudes. At 1.5 km, the AOD values remain notable over southern Spain and the western Mediterranean, but are much weaker compared to earlier days. At 3 km, the dust plume is localized over the Iberian Peninsula, with faint traces extending towards southeastern Europe. Higher altitudes, represented by $AOD_5$ and $AOD_{10}$, exhibit very low concentrations, indicating minimal vertical transport of the dust on the final day of the event. This vertical behaviour, compared to earlier days, reflects the dissipation phase of the Saharan dust plume, as atmospheric processes like mixing, dilution, and deposition progressively weaken its intensity. Additionally, during the Saharan dust event, sub-hourly estimated multi-layer AOD values for the European continent (from 07:12 to 16:12 daily on March 18, 2022) were compiled into an animation that includes 37 SEVIRI scenes over Europe, as shown in Pashayi et al., 2025 (https://doi.org/10.5446/69730).

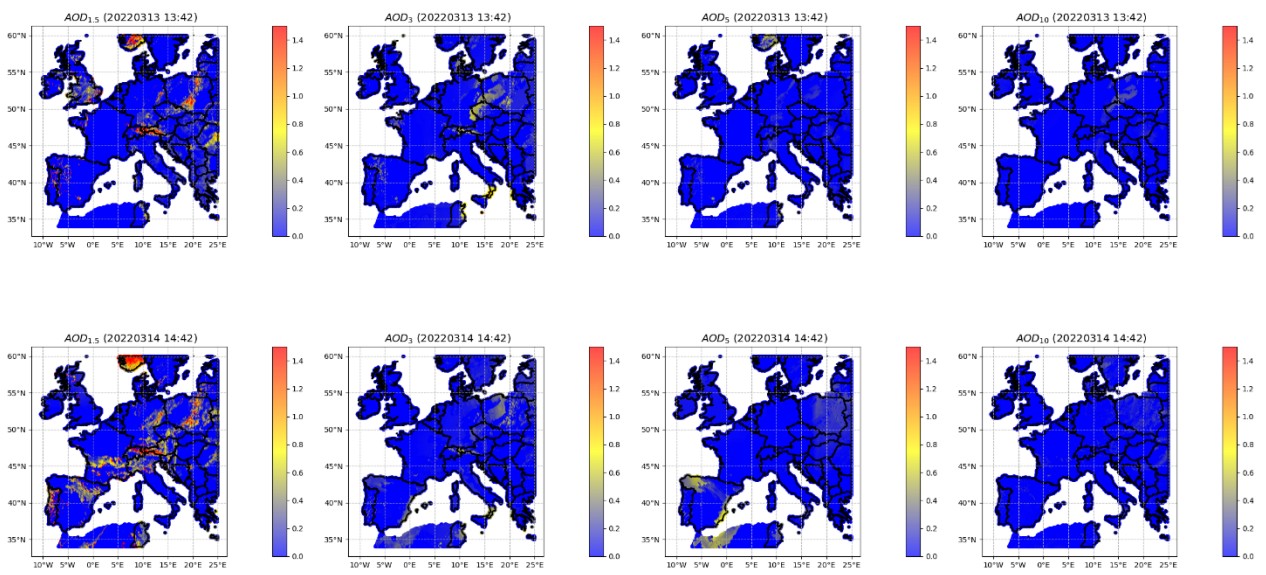

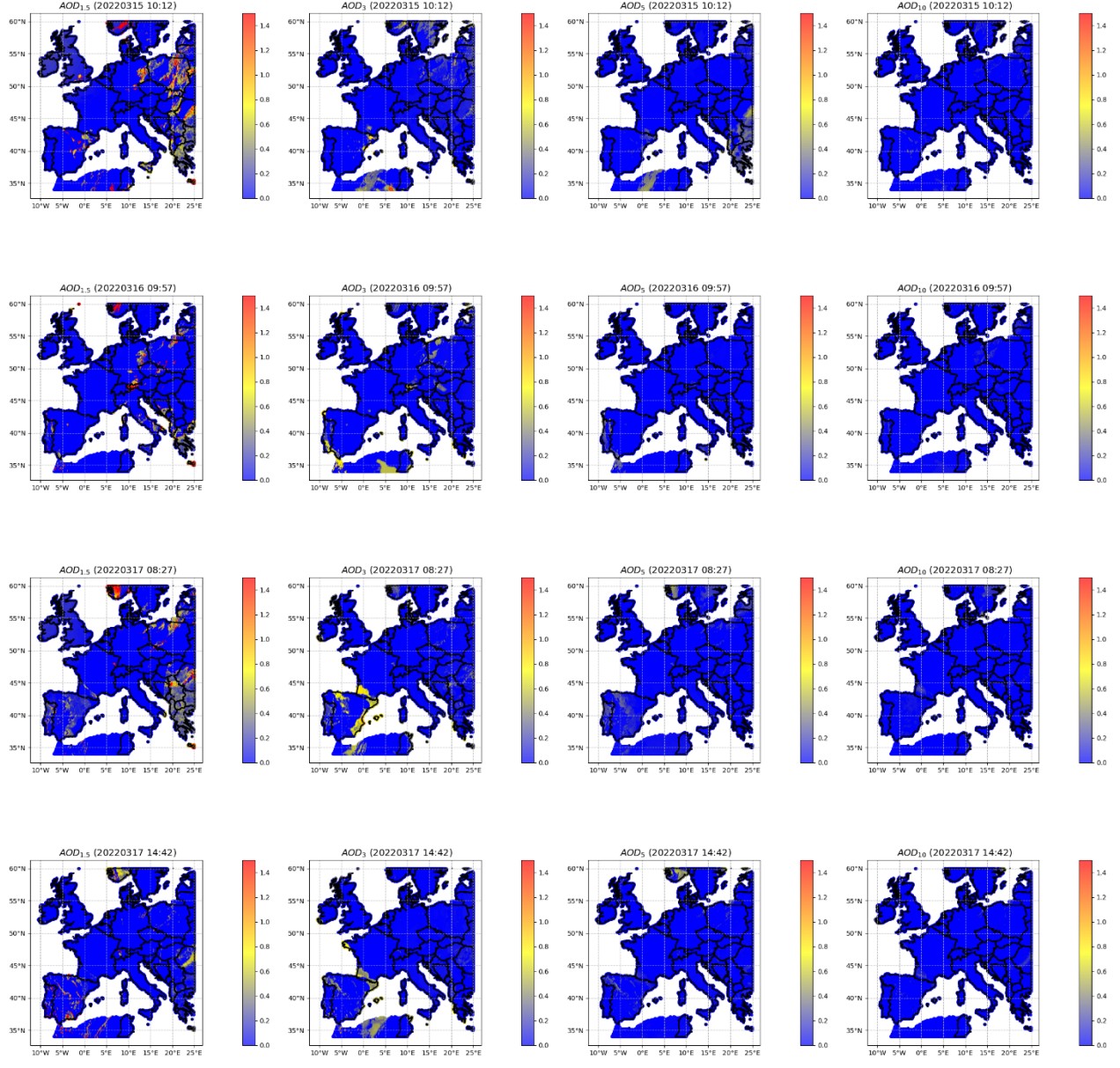

**Figure. 10. Spatial distributions of SEVIRI- estimated multi-layer AOD values during the Saharan dust event from March 13 to 18, 2022, at selected times.**

The other event in this approach is the Mount Etna eruption, located on the eastern coast of Sicily, Italy, one of the most active volcanoes in the world and a prominent feature in the Mediterranean region. On August 14, 2023, Mount Etna erupted in a significant volcanic event that released vast quantities of ash and aerosols into the atmosphere. This eruption produced an ash plume reaching up to 8,200 meters above the crater and spreading southward over the Mediterranean. The event provided

another opportunity to visualize the aerosol vertical dispersion behaviour aligned with the dynamic characteristics of the event. The AOD spatial distribution maps in Fig.11 revealed distinct layers of aerosol, with a clear upward transport of particles, especially in the lower to mid-level layers (1.5 km to 3 km) at 7:27, suggesting strong vertical convective activity driven by the eruption's intensity. As time progressed, the AOD values in the upper layers increased, particularly at the 5 km altitude

5    around 8:27, signalling a greater vertical transport of aerosols. This upward transport continued throughout the day, with the highest values observed at 10 km by 9:57, suggesting the plume had reached greater altitudes. By 13:42, the AOD values in the 10 km layer were at their peak, reflecting the maximum extent of vertical transport before the aerosols began to disperse more horizontally. After this point, the plume's vertical extent started to decrease (15:57), likely due to the long-distance transport of the aerosols across the Mediterranean region. This multi-layer analysis highlights the dynamic behaviour of

10   volcanic aerosols, providing valuable insight into their dispersal patterns. An animation of sub-hourly SEVIRI multi-layer AOD estimations during the eruption (from 07:27 to 15:57) was also generated, represented as Video in Pashayi et al., 2025 (https://doi.org/10.5446/69731), providing a detailed view of the plume's evolution over time.

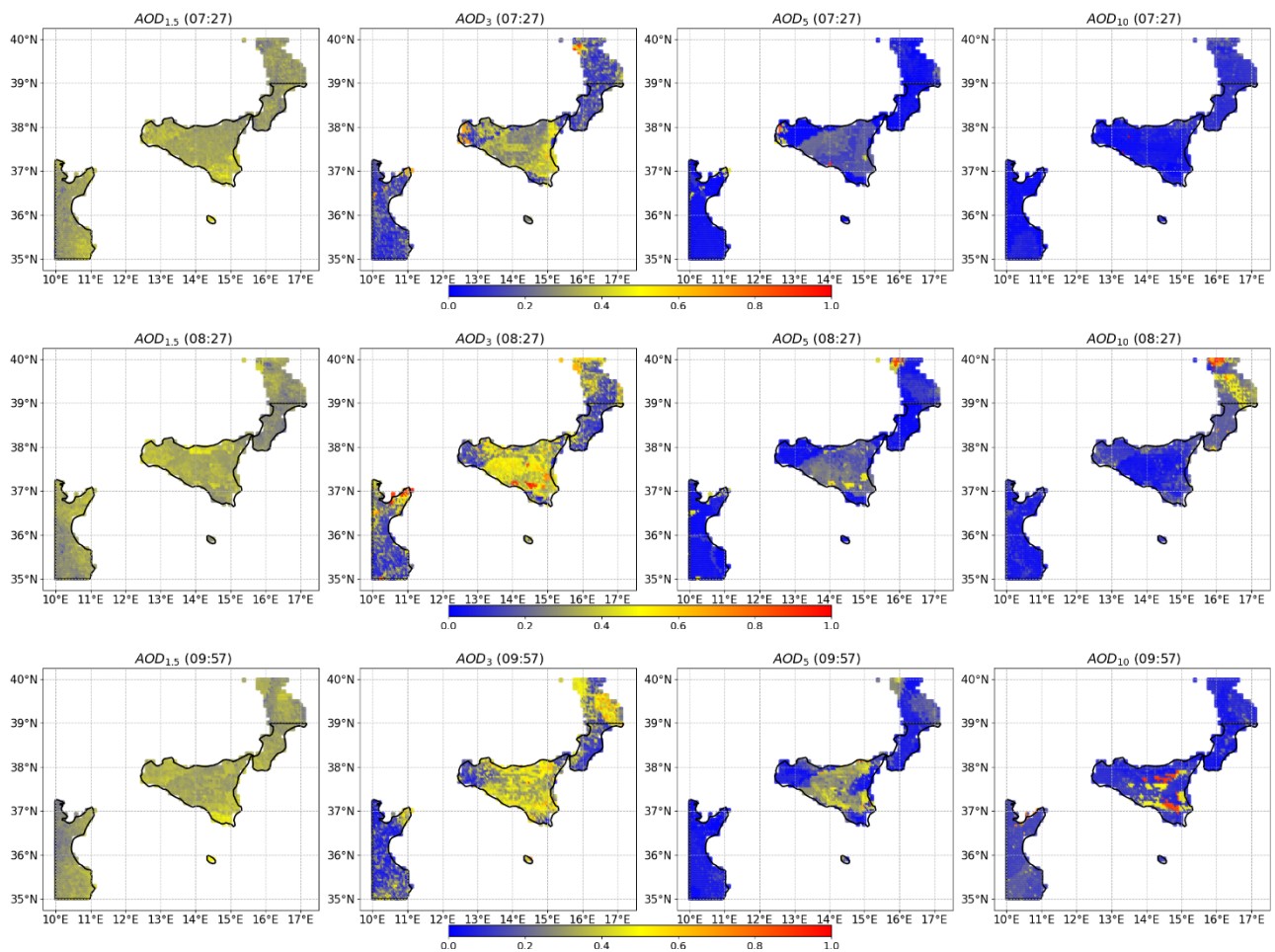

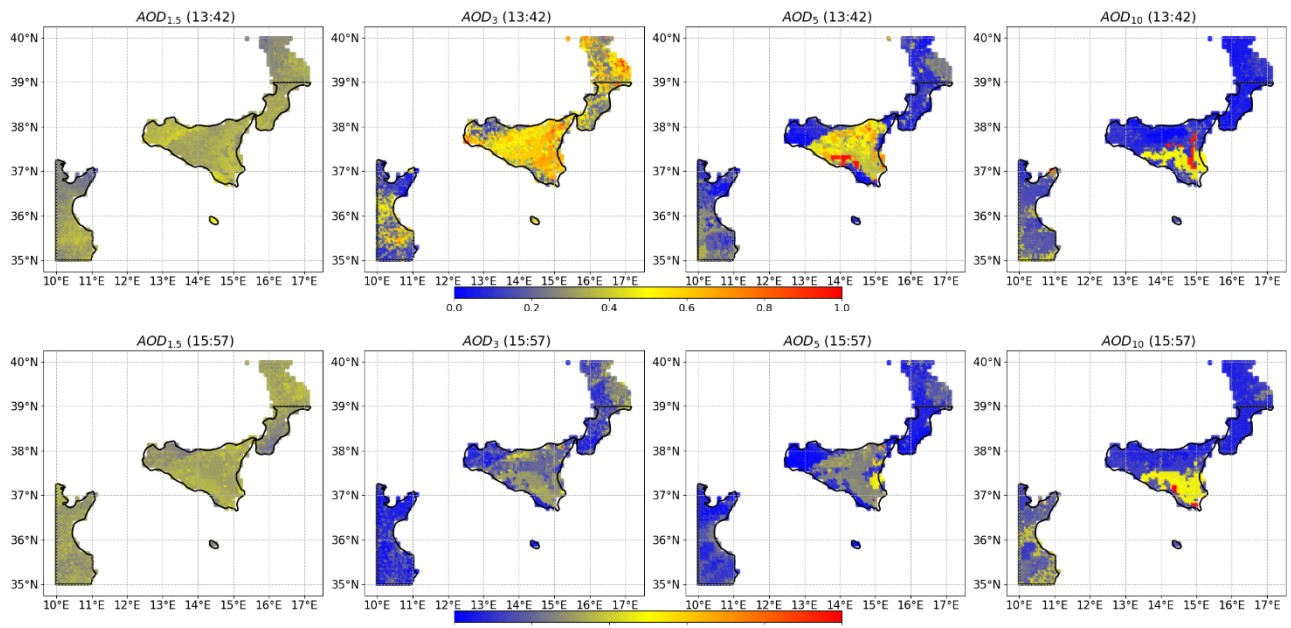

**Figure. 11. Spatial distributions of estimated multi-layer AOD values during the Volcanic Eruption on August 14, 2023, at selected times.**

**5 Conclusion**

This study proposes a model that integrates satellite TOA reflectance data from the SEVIRI sensor, meteorological data, and land cover to estimate AOD across distinct altitude layers at 1.5, 3, 5, and 10 km. Using CALIOP AOD profiles as reference data, RF and XGB models were trained on a dataset spanning 2017 to 2018. The trained models were then used to estimate

SEVIRI-based multi-layer AOD values over Europe for 2019 and 2020. The 2019 estimates were compared with CALIOP AOD retrievals, while the 2020 estimates were evaluated against EARLINET AOD retrievals, yielding the following insights. Both RF and XGB models demonstrate high accuracy in estimating sub-hourly SEVIRI multi-layer AOD values at approximately 15-minute intervals. The XGB model shows slightly superior performance, achieving $R^2$ values ranging from 0.97 to 0.99 across different layers when compared to CALIOP, and from 0.59 to 0.87 when evaluated against EARLINET

retrievals. Incorporating meteorological data such as T, P, Ws, and Wd, along with LC data, during model training significantly enhances the performance of the proposed frameworks. These additional features, often excluded in traditional physical AOD retrieval methods that rely solely on atmospheric radiative transfer models, greatly improve the accuracy of SEVIRI multi-layer AOD estimations. Among the meteorological variables, Ws and Wd are the most influential, resulting in higher $R^2$ values and lower RMSE across all estimated layers.

A qualitative validation was conducted by comparing the spatial trends of the estimated AOD values with CALIOP AOD retrievals. The analysis focused on SEVIRI pixels corresponding to CALIOP overpasses on March 3, April 30, June 13, and

October 31, 2019, with temporal differences of less than four minutes. The results demonstrate a strong agreement between SEVIRI estimates and CALIOP retrievals across varying AOD levels, highlighting the model's capability to provide reliable high-resolution AOD estimates that effectively complement CALIOP data. Additionally, the study successfully estimated multi-layer AOD at 15-minutes intervals for two real events —a Saharan dust plume that swept across Western and Central

Europe between March 13 and 18, 2022 and the Mount Etna eruption on August 14, 2023. The results are consistent with the physical characteristics of these phenomena, such as Saharan dust long-range transport in the upper layers of the atmosphere and a gradual increase in AOD values over time from the lower to higher tropospheric layers during volcanic events. This approach enables detailed monitoring of aerosol behaviour across vertical layers of the troposphere, providing valuable insights into the dynamics of such events. In conclusion the XGB model can estimate detailed sub-hourly 3x3 km² multi-layer AOD

values, providing valuable insights into aerosol properties.

Our research, cantered on the troposphere over Europe and validated with ground and satellite LiDAR-based AOD retrievals, provides a foundation for future studies to develop a more comprehensive approach for multi-layer AOD retrievals by incorporating an ensemble of geostationary meteorological satellites. Moreover, the current framework utilizes a limited range of input features, omitting important variables such as precipitation, NDVI, and land use, which significantly affect AOD

dynamics. Future efforts will focus on improving model accuracy by including these additional factors and exploring the potential applications of the model's outputs in areas like aerosol transport analysis, air quality assessment, and climate studies to enhance its practical relevance.

**Data and Material Availability.** Data will be made available upon request.

**Code availability.** Code will be made available on request.

**Author Contributions.** MP conducted the investigation, design, data curation, and processing, as well as the programming and evaluation. MP also wrote the original draft. MS was responsible for conceptualization, methodology, supervision, and validation, and contributed to the review and editing of the manuscript. MMSh contributed to the conceptualization, supervision, and validation of the work. All authors have read and approved the published version of the manuscript.

**Competing interests.** The authors declare that they have no conflict of interest.

**Acknowledgment.** The authors extend their gratitude to the SEVIRI team (https://eoportal.eumetsat.int) for their dedicated efforts in developing and continuously improving SEVIRI data products. They also acknowledge the MODIS Science Team for their invaluable work in processing and making available the Collection 6 MODIS Land Cover Dynamics (MCD12Q2) Product (www.ladsweb.modaps.eosdis.nasa.gov). We also extend our heartfelt thanks to the CALIOP

(www.eosweb.larc.nasa.gov) and EARLINET (https://data.earlinet.org/earlinet/) Profile Products for their crucial contribution as benchmarks, validating our model outputs and providing invaluable reference data. Additionally, we express our gratitude to the European Centre for Medium-Range Weather Forecasts (ECMWF) for granting us access to their meteorological dataset (https://cds.climate.copernicus.eu), which greatly enhanced our understanding and modeling of atmospheric conditions.

**Funding.** The authors declare that they have no known competing financial interests or personal relationships that could have

appeared to influence the work reported in this paper.

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
