# Peer review of "Multi-layer Retrieval of Aerosol Optical Depth in the Troposphere Using SEVIRI Data, Case study: European Continent"

_Atmospheric Measurement Techniques, 2024_

## Author Comment (AC1)

**Reviewer #2:**

➢ This paper presents an awkward title, "Vertical Retrieval of AOD...". There seems to be a misunderstanding regarding the term "vertical retrieval." AOD (Aerosol Optical Depth) is generally understood as a columnar quantity, representing the total extinction of light from the surface to the top of the atmosphere. The concept of "vertical retrieval" is unclear. Upon reading the paper, it appears that the authors are referring to the retrieval of AOD in layers at specific altitudes, with each layer having a thickness of 60 m, which corresponds to the vertical resolution of CALIOP data. If this is the case, a more appropriate title would be something like "Retrieval of Extinction at Four Layers Using...".

We sincerely thank you for dedicating your time to review our paper and providing valuable feedback. Your comment regarding the title and the term "vertical retrieval" is particularly insightful, and we appreciate the opportunity to clarify and address this concern.

Indeed, as you noted, the focus of our study is on estimating columnar AOD values within specific altitude layers: 0–1.5 km, 1.5–3 km, 3–5 km, and 5–10 km. We understand how the original title might have led to ambiguity, potentially suggesting retrievals at a finer vertical resolution or full vertical profiles.

To resolve this and better reflect the scope of our work, we have revised the title of the manuscript to "Multi-layer Retrieval of Aerosol Optical Depth in the Troposphere Using SEVIRI Data: A Case Study over the European Continent." Additionally, throughout the manuscript, we have replaced the term "SEVIRI AOD profiles" with "SEVIRI multi-layer AOD values" where applicable to ensure consistency and clarity.

We believe these changes effectively address your comment and improve the overall precision of the manuscript.

➢ From what I gather, the paper does not conduct a true retrieval of the aerosol extinction profile, as provided by CALIOP. If they are only retrieving AOD for a few layers, it raises the question—what is the purpose of this retrieval? Who will use such data? most climate models have vertical resolution coarser than 500m. so, layered AOD at 60 m at a few altitudes seems not useful.

You raise an important point regarding the nature of our manuscript's approach, and we appreciate your thoughtful feedback. In response to your question about the significance of AOD retrieval at specific altitudes (1.5 km, 3 km, 5 km, and 10 km) with high temporal resolution about 15 minutes, we have revised the Introduction section to better clarify the purpose and relevance of this approach as follows (Page 4 line 27 to Page 5 line 5):

"In this study, we introduce a model for sub-hourly multi-layer AOD retrieval over Europe continent troposphere by integrating SEVIRI-based information with CALIOP aerosol profile products. To achieve this, two well-established machine learning models—XGBoost (XGB) and Random Forest (RF)— utilized for retrieving AOD values in four distinct layers, approximately every 15 minutes, with a spatial resolution of 3 km × 3 km. The four tropospheric layers analyzed in this study are 0–1.5 km, 1.5–3 km, 3–5 km, and 5–10 km, denoted as $AOD_{1.5}$, $AOD_3$, $AOD_5$, and $AOD_{10}$, respectively. The selection of these layers for multi-layer AOD retrieval is based on the distinct aerosol transport mechanisms observed at these altitudes. The 0-1.5 km layer captures aerosols from local sources transported upwards

by updrafts from the cloud base, a process called pumping. The 1.5-3 km layer, where thermal bubbles often initiate, allows examination of aerosols, potentially from mid-range sources, that are lifted into the cloud with the rising bubble. The 3-5 km layer captures aerosols transported over longer distances that enter the cloud through entrainment at the cloud edges as the bubble ascends. The 5-10 km layer is designed to capture the influence of long-range transported aerosols on cloud properties at higher altitudes. This multi-layer approach enables analysis of how local to long-range aerosol transport contributes to aerosol-cloud interactions (Zhang et al., 2021; Lebo, 2014; Marinescu et al., 2017)."

Due to significant flaws in the methodology and validation, I recommend rejecting this paper. Below are my major comments:

➢ **Introduction**: The introduction fails to acknowledge recent advancements in aerosol layer height retrieval from instruments like EPIC and TROPOMI. I recommend referencing recent literature, such as https://doi.org/10.1016/j.rse.2021.112674 and the references therein.

> We sincerely thank you for introducing these valuable references to our attention regarding recent advancements in aerosol layer height retrieval, particularly from instruments like EPIC and TROPOMI. We have reviewed these studies thoroughly and have incorporated them into our literature review, considering their objectives, key achievements, and limitations in the context of our work.
> In response to your suggestion, we have revised the Introduction section to include a discussion of these recent advancements, ensuring that our manuscript aligns with the latest developments in the field of aerosol layer height retrieval (Page 2 line 23 to Page 3 line 25).

"Recent advancements have sought to overcome these limitations through the use of passive satellite sensors with varying temporal resolutions, such as the Tropospheric Monitoring Instrument (TROPOMI), which provide near-daily global coverage with a spatial resolution $3.5 \times 7$ km (improved $3.5 \times 5.5$ km in 2019) and was launched in 2017 on Sentinel-5P satellite (Veefkind et al., 2012); the Earth Polychromatic Imaging Camera (EPIC), offering a continuous daytime view every 60 to 100 minutes with a spatial resolution of about 8x8km since its lunch on February 11, 2015, onboard the Deep Space Climate Observatory (DSCOVR) satellite (Marshak & Knyazikhin, 2017); the Global Ozone Monitoring Experiment-2 (GOME-2) on Meteorological Operational Satellite Program (MetOp-C), with a three-day revisit cycle and a spatial resolution of approximately $40 \times 40$ km since 2018; and the Moderate Resolution Imaging Spectroradiometer (MODIS), onboard Terra (launched in 1999) and Aqua (launched in 2002), provides daily global coverage with spatial resolutions ranging from 0.25 to 1 km (Lyapustin et al., 2011).

The relevant researches focus on various methods specifically aimed at retrieving aerosol layer height (ALH) rather than AOD at different altitudes. One prominent method, Oxygen ($O_2$) A and B band Absorption Spectroscopy, utilizes the differential absorption of sunlight by $O_2$ molecules at different altitudes (Zeng et al., 2018; Xu et al., 2017; Xu et al., 2019). Elevated aerosol layers scatter sunlight back to space, shortening the atmospheric path length and decreasing $O_2$ absorption. By analyzing spectral characteristics in the $O_2$ A and B bands, researchers infer ALH. However, retrieval sensitivity is enhanced over darker surfaces and higher AOD, making it challenging over bright surfaces or under low aerosol loading. For instance, Nanda et al. (2020) employed TROPOMI observations with an optimal estimation scheme in the $O_2$ A band, assuming a uniformly distributed aerosol layer. Similarly, the algorithm

developed using EPIC/DSCOVR data leverages atmospheric window bands and Differential Optical Absorption Spectroscopy (DOAS) ratios, integrating MODIS and GOME-2 surface reflectance data. For retrievals over vegetated areas, the algorithm favours the $O_2$ B band due to its lower surface reflectance (Xu et al., 2019). Another study combined $O_2$ A and B band data from Scanning Imaging Absorption Spectrometer for Atmospheric Chartography (SCIAMACHY) and GOME-2 for enhanced ALH sensitivity, especially near boundary layers (Hollstein & Fischer, 2014).

Additional retrieval method, Stereoscopic techniques—employed by the Multi-angle Imaging SpectroRadiometer (MISR), launched in 2000— utilize multi-angle observations to geometrically determine plume heights. MISR offers a spatial resolution of approximately 275 meters and a temporal resolution of around once every 7 days, making it especially useful over reflective surfaces, as it relies on geometric data rather than surface reflectance (Muller et al., 2002; Zakšek et al., 2013; Fisher et al., 2014; Val Martin et al., 2018).

Passive satellite-based ALH retrieval techniques, while offering global coverage, often simplify the aerosol vertical distribution by assuming a single homogeneous layer (Zeng et al., 2018; Xu et al., 2017; Xu et al., 2019). This simplification can lead to inaccurate representations of complex aerosol profiles, especially in cases of multi-layered events. In addition, these passive satellite-based methods face further constraints due to the low spatial resolution of instruments like EPIC and GOME-2, as well as low temporal resolution of sensors such as TROPOMI, GOME-2, and MISR. These constrains on resolution reduce the effectiveness of these retrievals in capturing fine-scale, rapidly evolving aerosol distribution events, such as smoke plumes from fires.”

- ➢ **Methodology**: The method has significant flaws, particularly in the use of machine learning as a "black box." The authors fail to explain the underlying physics of how SEVIRI would contain information about the aerosol vertical distribution—specifically, at what wavelengths and why? The method pairs CALIOP layered AOD with meteorological data and SEVIRI radiance for training but lacks a clear justification for this approach.

Thank you for your valuable suggestion, which we believe greatly enhances the clarity and depth of our methodology. We have carefully considered your feedback and have revised the Methodology section to address your concerns. To provide a clearer understanding of how SEVIRI data contributes to the retrieval of aerosol vertical distribution, we have added several sentences at the beginning of Section 3. Specifically, we now discuss the impact of aerosols at the relevant altitudes and how different SEVIRI bands, meteorological data, and land cover types provide critical information for the machine learning models. Additionally, we have highlighted the importance of spatial and temporal features such as latitude, longitude, year, month, and day, and how these factors influence the retrieval process. These revisions help clarify the rationale behind pairing CALIOP layered AOD with SEVIRI radiance and meteorological data, providing a more robust justification for this approach (Page 9 line 2 to Page 10 line 30).

[revised manuscript text omitted]

➢ **Cross-validation**: The cross-validation approach is problematic. The data used for training and validation likely have significant auto-correlation in space or time. I suggest separating the datasets by year, for example, using the first two years of data for training and the third year for validation.

Thank you for your insightful suggestion regarding the cross-validation approach. We fully agree that ensuring temporal independence between training and testing datasets is crucial for avoiding potential issues related to autocorrelation. To address this, we have revised our data partitioning strategy to ensure better temporal generalization of the models. Specifically, the dataset was split by year, with the data from 2017 and 2018 used for model training, while the 2019 data was reserved exclusively for testing. This approach helps mitigate temporal autocorrelation and ensures more robust model evaluation.

In line with this updated approach, we have thoroughly revised the "3.3 Model Training and Evaluation" subsection. Additionally, all figures (Figures 6, 7, 8) and tables (Tables 3, 4, S1, S2) representing the results of the trained and tested models have been updated accordingly, and the related text has been modified to reflect these changes.

➢ **Validation and Results**: The paper should provide a map of retrieved AOD at each hour and validate these results with AERONET sites. Additionally, the paper should present an extinction profile from their retrieval and compare it with either ground-based lidar measurements or other aerosol profile measurements, such as those from aircraft.

Thank you for your valuable suggestion regarding the validation and results. To address the ambiguity related to the retrieval of profiles, we have revised the manuscript to clarify that we are estimating multi-layer AOD values rather than retrieving full aerosol extinction profiles. This change has been applied throughout the manuscript to ensure consistency and clarity.

Regarding the validation aspect, we acknowledge the importance of using reliable datasets. However, due to the nature of our approach—multi-layer AOD values estimation—AERONET station measurements cannot be comprehensively utilized for validation across all altitude layers. Instead, we have validated our results using EARLINET ground-based lidar AOD retrievals for the four specified altitude ranges. These validations are presented in Figure 8 and Table 4 (Page 24 line 1 to Page 27 line 1).

For further validation, we have conducted a detailed spatial comparison of the estimated multi-layer AOD values with CALIOP AOD retrievals on four specific days across different seasons in 2019, as illustrated in Fig. 9 (Page 27 line 1 to page 29 line 1). Additionally, to enhance both temporal and spatial validation, we have included two significant

aerosol events: a major dust intrusion and a volcanic eruption. These events are analyzed in detail and presented in Fig. 10, Fig. 11, Video S1, and Video S2 (Page 29 line 1 to 33 line 1). The spatial and temporal exploration of these events provides additional validation of the estimated multi-layer AOD values, further reinforcing the robustness of the proposed methodology.

- We sincerely hope that these revisions address your concerns and enhance the clarity and overall impact of the manuscript. We greatly appreciate your time and thoughtful feedback, which have been instrumental in refining our work. Thank you once again, Reviewer 2, for your invaluable comments and suggestions.

---

## Author Comment (AC2)

**Reviewer #1:**

➢ This is a review for "Vertical Retrieval of AOD using SEVIRI data, Case Study: European Continent" by Pashayi et al. In this paper, AOD is determined at several altitudes from SEVIRI/MSG data using machine learning (ML) methods trained on vertical aerosol profiles from the CALIOP lidar satellite. Validation is performed using CALIOP and independent EARLINET ground lidar data. The article is well written and generally clear.

We sincerely appreciate your thoughtful and constructive feedback on our manuscript titled *"Vertical Retrieval of AOD using SEVIRI Data: A Case Study over the European Continent."* We are grateful for the time and effort you dedicated to reviewing our work and for your positive assessment of its clarity and presentation.

Your insights and suggestions are invaluable to us, and we are committed to addressing your comments thoroughly to further enhance the quality and scientific rigor of our manuscript. We greatly value your expertise and guidance in helping us refine our research for publication.

➢ However, I find the scientific interest of the study to be very limited, at least in the way findings are presented. Indeed, results are limited to a large number of scores calculated by validating the vertical variation of the AOD retrieved at certain single pixel locations, thus showing no maps or vertical profiles estimated by the presented methodology.

Thank you for your insightful suggestion emphasizing the importance of incorporating spatial and temporal visualizations of the retrieved multi-layer AOD values to better demonstrate the applicability of our methodology. In response to your feedback, we have added a new subsection, *"4.3 Qualitative Validation,"* which includes the following enhancements (Page 27 line 1 to Page 33 line 1):

1. **Spatial Maps**: We have incorporated maps illustrating the spatial trends of the multi-layer AOD retrievals and their comparison with CALIOP AOD retrievals on four specific days across various seasons in 2019.

2. **Temporal Analysis**: We have conducted and presented detailed analyses of two significant aerosol events: (i) a major dust intrusion event from March 13–18, 2022, and (ii) a volcanic eruption event on August 14, 2023, as the sub-hourly estimated multi-layer AOD values.

These additions provide illustrative examples that highlight the spatial and temporal capabilities of our proposed methodology. We believe these enhancements substantially improve the manuscript and effectively address your concerns.

➢ Furthermore, the temporal variation (daily and diurnal) of the retrieved aerosol variables is not discussed, nor shown, in the paper. I find this highly surprising, mainly because the authors state in the abstract that "These estimations are achieved with spatial and temporal resolutions of 3 km × 3 km and 15 minutes, respectively, over Europe". In my opinion, this statement needs to be proven in order to properly evaluate the contribution of this paper, as SEVIRI's main asset is that it is a high temporal resolution imager that provides images of the entire disk of the Earth every 15 minutes.

Thank you for your valuable feedback. We have carefully considered your suggestion regarding the inclusion of the temporal variation of the estimated multi-layer AOD values. In response, we conducted a detailed regional-scale analysis focusing on two significant aerosol events: (i) a major dust intrusion from March 13 to March 18, 2022, and (ii) a volcanic eruption on August 14, 2023. To demonstrate the spatial and temporal capabilities of the proposed methodology, we have included spatial distribution maps at key time intervals during these events. These maps effectively illustrate the temporal evolution of aerosols across four altitude layers. This analysis, along with the corresponding visualizations, has been incorporated into the manuscript as *Figures 10 and 11* within the new subsection *"4.3 Qualitative Validation."*. Additionally, we have compiled sub-hourly estimations of multi-layer AOD values into an animation, provided as Supplementary Video S1 and S2, which visualizes how aerosols dynamically evolve over time. We believe these additions address your concerns and further substantiate the temporal capabilities of our methodology (Page 29 line 1 to Page 33 line 1).

[revised manuscript text omitted]

I suggest including examples of maps and time series of vertical AOD variations across space and time, as well as their validation with reference data, as this is in my eyes the main interest of the proposed methodology for the scientific community.

We greatly appreciate your suggestion to include spatial and temporal validation to better highlight the main scientific interest of the proposed methodology. In response, we have incorporated maps under the subsection *"4.3 Qualitative Validation"* that demonstrate the spatial trends of estimated multi-layer AOD values and their comparison with CALIOP AOD retrievals. These maps provide a visual means for readers to assess the consistency between the estimated multi-layer AOD values and the CALIOP reference data, thereby strengthening the validation and relevance of our methodology (Page 27 line 1 to Page 29 line 1).

"Conducting the qualitative validation for an entire scene within the first approach is challenging due to the spatial and temporal resolution constraints of CALIOP. To address this limitation, SEVIRI scene pixels corresponding to CALIOP overpasses with temporal differences of less than four minutes were compared on the specified days. The results, illustrated in Fig. 9, indicate that the spatial trends of the estimations generally align well with the trends of CALIOP AOD retrievals in regions with both high and low AOD values across the four seasons. This alignment highlights the model's ability to provide reliable AOD estimates with enhanced temporal resolution, effectively complementing CALIOP AOD retrievals.

[Figure]

[Figure]

Figure. 9. Spatial distributions and trends of SEVIRI-retrieved multi-layer AOD values compared to CALIOP retrievals for four specific days in March 3 (11:57), April 30 (12:42), June 13 (10:57), and October 31 (12:27) across various seasons in 2019.

➢ I strongly suggest going beyond the statistical analysis, which is extensively presented in this paper, to discuss more the physical interpretation of the results. This applies not only to the results in general but also to other parts of the paper, such as the analysis of the importance of different variables in vertical AOD retrieval.

Thank you for your valuable feedback. While statistical modeling provides important insights, we acknowledge the need to complement these analyses with a thorough understanding of the underlying meteorological and physical

mechanisms. In response, we have revised Sections *"4.1.1"* and *"4.1.2"* to include a more comprehensive physical interpretation of the results.

These revisions emphasize the importance of different variables and their contributions to multi-layer AOD retrievals, offering a more holistic perspective on the physical relevance of key features identified by the machine learning models. Specifically, we discuss the sensitivities of SEVIRI spectral bands, the influence of meteorological parameters (e.g., wind speed and humidity), and the role of land cover types. Additionally, we highlight how these factors shape the model's performance, extending the analysis beyond statistical metrics to capture the physical underpinnings of the retrieval process (Page 16 line 9 to Page 20).

[revised manuscript text omitted]

➢ To avoid writing a lengthy paper, some details about the methodology can be omitted (for example, I would only consider the RF method if it is the best of the two).

We appreciate your suggestion to streamline the methodology section for improved clarity and conciseness. While we acknowledge the merit of focusing on the best-performing model, we believe it is important to retain both models (RF and XGB) in the manuscript. To address this, we have reduced the detailed descriptions of model architectures and hyperparameter tuning, concentrating instead on their comparative performance. To further streamline the presentation and reduce redundancy, we have omitted Table 4 and Table S3, as their content is effectively covered through other figures and analyses. Including both models allows us to provide a more comprehensive evaluation, demonstrating the robustness of the methodology across different approaches. This comparison also highlights the adaptability of the proposed methodology in varying contexts, which we believe is valuable for the scientific community. By balancing detail with conciseness, we have ensured that the manuscript remains accessible while maintaining its depth and rigor.

▪ We sincerely hope that these revisions address your concerns and improve the clarity and impact of the manuscript. We thank you again for your time and insightful comments, which have been invaluable in refining our work.